# Bags of Projected Nearest Neighbours: Competitors to Random Forests?

**David P. Hofmeyr**                                                       *d.p.hofmeyr@lancaster.ac.uk*
*School of Mathematical Sciences*
*Lancaster University, United Kingdom*

**Reviewed on OpenReview:** *https://openreview.net/forum?id=ZKLj2UOCsO*

## Abstract

In this paper we introduce a simple and intuitive adaptive $k$ nearest neighbours classifier, and explore its utility within the context of bootstrap aggregating ("bagging"). The approach is based on finding discriminant subspaces which are computationally efficient to compute, and are motivated by enhancing the discrimination of classes through nearest neighbour classifiers. This adaptiveness promotes diversity of the individual classifiers fit across different bootstrap samples, and so further leverages the variance reducing effect of bagging. Extensive experimental results are presented documenting the strong performance of the proposed approach in comparison with Random Forest classifiers, as well as other nearest neighbours based ensembles from the literature, plus other relevant benchmarks. Code to implement the proposed approach is available in the form of an `R` package from `https://github.com/DavidHofmeyr/BOPNN`.

## 1    Introduction

Bootstrap aggregating (Breiman, 1996), or "bagging", is the approach of combining the outputs of several predictive models, each fit to different bootstrap samples from a set of data, into a single *ensemble* predictive model. Bagging has remarkable potential for improving the prediction performance of high variance predictors, due to the variance reducing effect of model averaging. However, it is well understood that not all high variance predictors are able to leverage this effect equally, due to some being "too stable" across different bootstrap samples. Bagged ensembles of Decision Tree (DT) based models have undeniably shown the greatest promise to date, to the extent that bagging is sometimes categorised as a decision tree based approach (Hastie, 2009). The remarkable success of Random Forest (Breiman, 2001, RF) based models has only further entrenched DTs as the *de facto* "optimally baggable" model. Trees in RFs differ from regular DTs only through the addition of a randomisation step preceding each stage in the standard Classification And Regression Trees (Breiman, 2017, CART) algorithm. However, this simple modification has a remarkable "destabilising" effect on the already highly variable trees, and so enables further variance reduction through averaging.

Attempts have been made to emulate the success of RFs and bagged DTs with other non-parametric models, such as those based on nearest neighbours (Zhou & Yu, 2005; Cannings & Samworth, 2017; Gul et al., 2018). However, it is questionable whether any of these approaches has the potential to be a real competitor to RFs across many different settings, partly due to limited experimental results having been documented.

A necessary condition for the success of a bagged ensemble is substantial diversity in the models (Krogh & Vedelsby, 1994). However, existing approaches for inducing this diversity in nearest neighbours (NN) based ensembles have largely been based on randomisation alone (Zhou & Yu, 2005; Domeniconi & Yan, 2004; Deegalla et al., 2022), possibly owing to the successful application of randomisation within RFs. But purely randomised methods can only be beneficial if the resulting increase in diversity across models substantially outweighs the decrease in accuracy of the individual models caused by the extra randomness. Where RFs are fundamentally advantaged over these NN ensembles, however, is in the adaptive way in which DTs

determine their "smoothing neighbourhoods", with the additional randomisation being a secondary factor. The adaptiveness of DTs simultaneously is a primary source of diversity in the models beyond the randomness of the bootstrap sampling, and also reduces the amount by which the added randomness of RFs impacts on the quality of the individual models.

In this paper we introduce an intuitive adaptive $k$ Nearest Neighbours ($k$NN) classifier which is computationally efficient to compute, and explore its utility within bagged ensembles. Our main objectives with this piece of work are (i) to illustrate the importance of, and potential offered by, including an adaptive learning step within each model in a bagged $k$NN ensemble; and (ii) to support this illustration with a rigorous and extensive set of experiments.

The remainder of this paper is organised as follows. In the next section we provide some background on bagging, with particular focus on its application to $k$NN based models. In Section 3 we describe our approach, as well as some of the practicalities surrounding implementation and useful outputs from the resulting models. In Section 4 we document the results from experiments using all 162 data sets in the Penn Machine Learning Benchmarks repository (Olson et al., 2017, PMLB). Here we compare the performance of the proposed approach with RFs, as well as numerous other models for context. RF classifiers are viewed by many as excellent general purpose models; seldom much worse than any others, and frequently among the best performing models on data from extremely diverse domains. In order to support any new model as a realistic alternative in this regard, there should therefore be no possibility of data set selection bias (whether conscious or unconscious) which is possible whenever any subset of available data sets without a clear and justifiable selection criterion is used.

## 2 Bagging, $k$NN, and What's Been Tried

In this section we provide light technical background on bootstrap aggregating, and discuss some of its applications to $k$NN based classifiers. Let $D := \{(\mathbf{x}_1, y_1), (\mathbf{x}_2, y_2), ..., (\mathbf{x}_n, y_n)\}$ be a sample of realisations from a distribution $F_{X,Y}$, on $\mathbb{R}^d \times [K]$, where $[K] = \{1, ..., K\}$. That is, the *response variables* (or "class labels"), $y_i; i \in [n]$, each takes on one of $K$ known and distinct values, and the associated observations of the *covariates*, $\mathbf{x}_i; i \in [n]$, are each $d$-dimensional real vectors. This *training* sample, $D$, is then used to fit a model, $g(\cdot|D)$, which is used to predict the class label for any given *query point*, $\mathbf{x} \in \mathbb{R}^d$.

Bagging operates by resampling from $D$ multiple times to produce $B$ *bootstrap* samples, $\tilde{D}_1, ..., \tilde{D}_B$, and then combining the resulting models, $g(\cdot|\tilde{D}_b); b = 1, ..., B$, to obtain a final predictive model. Note that whether each bootstrap sample is obtained by sampling with or without replacement often has relatively little impact on the performance of the overall model. For ease of exposition we assume sampling without replacement, and in such a way that each bootstrap sample contains $n_B = \lceil \pi_B n \rceil$ observations, where $\pi_B \in (0, 1)$.

### 2.1 Bagging and Variance Reduction

Bagging can be remarkably effective in reducing the variance of flexible predictive models. This is most conveniently communicated when combining the individual models through averaging, i.e., when using

$$g^{(bag)}(\mathbf{x}|D) = \frac{1}{B} \sum_{b=1}^{B} g(\mathbf{x}|\tilde{D}_b). \tag{1}$$

It is straightforward to show that (Hastie, 2009),

$$Var(g^{(bag)}(\mathbf{x}|D)) = Var(g(\mathbf{x}|\tilde{D}_1)) \left( \rho + \frac{1-\rho}{B} \right), \tag{2}$$

where $\rho = Cor(g(\mathbf{x}|\tilde{D}_1), g(\mathbf{x}|\tilde{D}_2))$. Note that although each $\tilde{D}_b; b \in [B]$, has the same distribution as an i.i.d.[1] sample of the same size drawn directly from the underlying population, it is not the case that the joint distributions of any pair $\tilde{D}_{b_1}, \tilde{D}_{b_2}; b_1 \neq b_2$ are the same as those of pairs of independent samples from

---

[1]recall that we consider resampling without replacement

the population due to the (potential) overlap of the bootstrap samples. It is this fact which results in $Cor(g(\mathbf{x}|\tilde{D}_1), g(\mathbf{x}|\tilde{D}_2))$ generally being greater than zero. Bagging is beneficial, therefore, when the bias and variance of $g(\mathbf{x}|\tilde{D}_1)$ are similar to those of $g(\mathbf{x}|D)$, i.e., using a smaller sample does not affect accuracy too substantially, *and* where $Cor(g(\mathbf{x}|\tilde{D}_1), g(\mathbf{x}|\tilde{D}_2))$ is relatively small.

Where bagging really shines is when applied to flexible, low-bias models, between which the correlation due to overlapping samples is relatively low. Generally speaking the class of non-parametric smoothing models can be made extremely flexible by selecting a small "smoothing parameter". For example, the $k$NN model bases its prediction for a point, $\mathbf{x}$, only on the properties of the nearest $k$ points to $\mathbf{x}$ from among the $\mathbf{x}_i; i \in [n]$. However, $k$NN, and other so-called "lazy learners", have been referred to as "too stable" from the point of view of bagging, because the correlation induced by overlapping bootstrap samples is substantial. This can be intuited by considering the *region of influence* of an observation, say $\mathbf{x}_i$, as the subset of $\mathbb{R}^d$ to which $\mathbf{x}_i$ is one of the $k$ nearest from among the sample. Note that this region is completely independent of the observations of the response variable, and may depend on only a very small number of other sample points. This independence of the responses means the standard $k$NN model is not able to leverage the relationships between the covariates and the response in order to improve its fit (hence the term "lazy learner"). The extreme localisation implied by the fact that the region of influence of a point depends on so few other points is also why the $k$NN predictions from two samples with substantial overlap are so correlated. Decision trees, on the other hand, are *adaptive* non-parametric smoothers, and aggressively exploit the relationships between the covariates and response in how they recursively split up the input space to actively determine the regions of influence of each point. In this way the region of influence of each point can be dependent on every other point in the sample. As a result the *non-overlapping parts* of two bootstrap samples are able to differentiate their respective models sufficiently to induce lower correlation between their predictions.

**Remark 1** *Although the intuition underlying the effectiveness of bagging is most easily communicated from a model averaging perspective, it is worth pointing out that directly averaging class labels is nonsensical. Nonetheless, the formulation in Eq. (1) applies to the classification context if we use one of the following formulations:*

1. *The outputs of the individual models, $g(\mathbf{x}|\tilde{D}_b); b \in [B]$, are estimates for the full conditional distribution of $Y|X = \mathbf{x}$. Taking the average of such outputs is therefore also an estimate for this conditional distribution, and classifying according to the mode of this distribution is Bayes optimal.*

2. *The outputs of the individual models are indicator vectors for the predicted class of $\mathbf{x}$. Averaging these indicator vectors and assigning the final classification using the mode is in this case equivalent to the "majority vote" rule. Here the quantity $g^{(bag)}(\mathbf{x}|D)$ is not an estimate for the distribution of $Y|X = \mathbf{x}$, but rather an estimate for the distribution of $g(\mathbf{x}|\tilde{D}_0)$, where $\tilde{D}_0$ is a sample of size $n_B$ drawn directly from the underlying population.*

## 2.2 Bagged $k$NN Classifiers

Although standard $k$NN models are seen to be stable from the point of view of bagging, there is a prevailing opinion that they can be "destabilised" by adding randomisation to the way in which the neighbours of each point are determined. This can be achieved in multiple ways, such as only computing distances on a (random) subset of the variables in $\mathbb{R}^d$ (Domeniconi & Yan, 2004); by randomly projecting the observations before computing distances (Deegalla et al., 2022); or by using a random selection of the value of $p$ within the $L_p$-norm derived distance function itself (Zhou & Yu, 2005). By modifying the distance metric a greater variety of points can have an impact on the regions of influence of others. However, none of these approaches is adaptive to the relationships between the covariates and the response, and there is insufficient evidence that purely randomised approaches are useful in general. Since the dominant term in $Var(g^{(bag)}(\mathbf{x}|D))$ is equal to $\rho Var(g(\mathbf{x}|\tilde{D}_1))$, a modification such as this can only be beneficial if the reduction in $\rho$ outweighs the increase in the variance of the individual models. However this may be substantial if the modification is purely random.

Nearest neighbours models can be made adaptive by actively learning a distance metric to enhance discrimination of classes, either globally (Goldberger et al., 2004) or locally (Hastie & Tibshirani, 1995). As far as

we are aware, however, no such approaches have been explored within the context of bagging, likely because of the computational demand of fitting a large number of such models. Somewhere between fully adaptive and randomised is the approach of selecting from among multiple $k$NN models arising from different random projections of the observations. This approach has shown success in the context of bagging (Cannings & Samworth, 2017), however the number of individual $k$NN models is equal to the product of the number of bootstrap samples and the number of random projections from among which to select each model in the ensemble. This results in a considerable computational restriction. A related method (Gul et al., 2018), which also uses a selection from multiple randomised $k$NN models works as follows. First a large collection of models is fit to different bootstrap samples, with each using its own random selection of the variables in $\mathbb{R}^d$. Then a fixed proportion of these is selected for inclusion in the ensemble. However, since the selection of each model in the final ensemble is from the same collection of candidates, the models in the ensemble are strongly dependent. To counteract this, the selection of models is not purely based on their apparent predictive ability, but is also made to ensure some level of diversity in the predictions across models. Although the total number of models to be fit is substantially fewer than when a fully independent selection is made for each model in the ensemble, this approach is still substantially slower than alternatives. Moreover, this approach loses the statistical "niceness" of bagging, and in particular does not provide any Out-Of-Bag (OOB) estimates for performance. This further limits its applicability when any substantial hyperparameter search is needed to obtain a good model.

## 3   Bagging Adaptive $k$NN Classifiers Based on Discriminant Projections

The benefits of adaptive learning (as applied in fitting DTs) within the context of bagging are at least two-fold: (i) it enables the models to exploit the relationships between the covariates and the response; and (ii) it allows the region of influence of a point to depend on the entire sample, and not only on points which are very local to it. This latter fact induces further differentiation across the outputs of models fit to different bootstrap samples, and so reduces their correlation.

Making $k$NN classifiers adaptive by optimising the distance metric used in the nearest neighbour search is intuitively pleasing, and in principle has the potential to achieve the same benefits as those described above for DTs. However, most adaptive $k$NN methods are considerably more computationally demanding than is fitting DTs. This limits their application within the computationally intensive bagging framework. What seems largely to have been overlooked, however, is that adaptively modifying the distance metric to enhance class discrimination does not have to be performed in a fully optimal manner in order to leverage the benefits mentioned above. We therefore explore this potential through a more computationally efficient alternative, based on finding discriminant subspaces designed to enhance class discrimination as determined by $k$NN classifiers.

### 3.1   A Simple Discriminant Subspace for $k$NN

Discriminant subspaces are subspaces of $\mathbb{R}^d$ within which the classes are (relatively) easily separated from one another. Depending on the classifier being applied after projection, different formulations of the discriminant subspace will be more/less appropriate. For example, the well known Linear Discriminant Analysis (Rao, 1948, LDA) model, in which each class is modelled with a Gaussian distribution and all classes share a common covariance matrix, has a so-called "sufficient subspace" given by the eigenvectors of $\hat{\Sigma}_w^{-1}\hat{\Sigma}_b$ associated with its non-zero eigenvalues. Here $\hat{\Sigma}_w$ is the pooled within-class covariance estimate and $\hat{\Sigma}_b = \hat{\Sigma} - \hat{\Sigma}_w$, where $\hat{\Sigma}$ is the overall data covariance. The more flexible Mixture Discriminant Analysis (Hastie & Tibshirani, 1996, MDA) models each class with a Gaussian mixture. If all mixture components across all classes share a common covariance matrix, then a similar discriminant subspace can be obtained. When more general formulations are adopted, discriminant subspaces can be obtained by maximising the a classification likelihood objective (Peltonen & Kaski, 2005), or by maximising some measure of divergence of the distribution with all classes combined from the average within class distribution (Zhu & Hastie, 2003), with densities estimated on the projected data. These latter approaches require numerical optimisation, and are thus not computationally competitive with those which can be obtained using highly optimised eigen-solvers.

Generally speaking, however, discriminant subspaces can be thought of as pushing points into high density regions within their own classes, and into low density regions within other classes; and the appropriateness of a subspace depends on how density is being modelled. Motivated by this simple but principled idea, we adopt the following heuristic, which has some similarity with a fully non-parametric MDA. For each $i$, we let $i_k$ be the $k$-th nearest observation to $\mathbf{x}_i$ from within its own class, and $i'_k$ the $k$-th nearest observation to $\mathbf{x}_i$ from among those in other classes. We then define

$$\hat{\Sigma}_{in} := \frac{1}{n}\sum_{i=1}^{n}(\mathbf{x}_i - \mathbf{x}_{i_k})(\mathbf{x}_i - \mathbf{x}_{i_k})^\top, \quad \hat{\Sigma}_{out} := \frac{1}{n}\sum_{i=1}^{n}(\mathbf{x}_i - \mathbf{x}_{i'_k})(\mathbf{x}_i - \mathbf{x}_{i'_k})^\top. \tag{3}$$

For a unit vector $\mathbf{u} \in \mathbb{R}^d; ||\mathbf{u}|| = 1$ the quantity $\mathbf{u}^\top\hat{\Sigma}_{in}\mathbf{u}$ (respectively $\mathbf{u}^\top\hat{\Sigma}_{out}\mathbf{u}$) is then the average squared distance from each point to its $k$-th *same class neighbour* (respectively *other class neighbour*), measured along direction $\mathbf{u}$. Such unit vectors which lead to small values of $\mathbf{u}^\top\hat{\Sigma}_{in}\mathbf{u}$ and large values of $\mathbf{u}^\top\hat{\Sigma}_{out}\mathbf{u}$ are thus desirable *discriminant directions* for a $k$NN classifier. A sensible discriminant subspace is therefore formed by simply taking the leading eigenvectors of $\hat{\Sigma}_{in}^{-1}\hat{\Sigma}_{out}$.

**Remark 2** *Although the quantities $\mathbf{u}^\top\mathbf{x}_{i_k}$ and $\mathbf{u}^\top\mathbf{x}_{i'_k}$ will tend not to be precisely the $k$-th nearest in- and out-of- class neighbours to $\mathbf{u}^\top\mathbf{x}_i$ (i.e., the ordering of distances changes after projection onto $\mathbf{u}$), they nonetheless tend to be from among the nearer in- and out-of-class points, and so minimising the post-projection in-class and maximising the post-projection out-of-class distances still has the desired effect.*

**Remark 3** *It is of course possible to combine $k$NN with alternative discriminant subspaces within a bagged ensemble. However, as described above, the appropriateness of a subspace for discriminating classes depends on how the classes are to be modelled within that subspace. A subspace focused on minimising the near neighbour in-class distances and maximising the near neighbour out-of-class distances is therefore better suited for classification with $k$NN than would be, for example, the LDA subspace, which is only based on the first and second order structure of the class distributions.*

**Remark 4** *The quantity $\hat{\Sigma}_{in}$ can be seen as capturing the average local within class covariance, and is similar to the average within component covariance matrix used in MDA with a very large number of components. A directly analogous subspace would thus arise from the eigenvectors of $\hat{\Sigma}_{in}^{-1}(\hat{\Sigma} - \hat{\Sigma}_{in})$. However, we have found substantially superior performance using the eigenvectors of $\hat{\Sigma}_{in}^{-1}\hat{\Sigma}_{out}$. We believe this is at least partly due to the fact that $\hat{\Sigma}_{in}^{-1}\hat{\Sigma}_{out}$ uses a greater amount of information from the sample than does $\hat{\Sigma}_{in}^{-1}(\hat{\Sigma} - \hat{\Sigma}_{in})$, and therefore intuitively leads to greater diversity across bootstrap samples.*

**Remark 5** *It is worth noting that we use only the $k$-th in- and out-of-class neighbours of each point when determining the discriminant subspace despite the fact that the classification of a point after its projection is based on all of its $k$ nearest neighbours. This is ultimately because the density, as quantified by $k$NN, is based only on the $k$-th neighbour distance. Perhaps more intuitively, minimising the $k$-th in-class neighbour distance has the effect of making the ball containing all $k$ nearest neighbours as small as possible.*

### 3.1.1 Additional Diversity Through Randomised Variable Selection

As described previously, additional randomness across different models (e.g. through randomised variable selection) can be beneficial if the resulting decrease in accuracy of the models arising from this randomness (which may also be seen as a reduction in information) is outweighed by the effect of additional diversification of the individual models. Adaptive models mitigate the reduction in accuracy arising from reduced information, when compared with lazy learners, as they better exploit what information *is* available and are better suited to filtering out noise.

In our preliminary experiments it became clear that restricting the discriminant subspaces each to lie within their own randomly determined higher dimensional subspace does indeed improve performance in general. How we achieve this is simply by only using information from a (random) subset of covariates in each model in an ensemble. Specifically, let $Q \subset [d]$ be a subset of the total covariates (which in practice is determined randomly), with $|Q| = q_0$, and let $\mathbf{I}_Q \in \mathbb{R}^{d \times q_0}$ have as columns the cardinal basis vectors for the

dimensions in $Q$. Then define $i_k$ and $i'_k$ as before, except with distances only computed with the variables in $Q$. We then define the *discriminant matrix*, whose eigenvectors define the discriminant subspace, simply by $\hat{\Delta} := \mathbf{I}_Q \left( \mathbf{I}_Q^\top \hat{\Sigma}_{in} \mathbf{I}_Q \right)^{-1} \left( \mathbf{I}_Q^\top \hat{\Sigma}_{out} \mathbf{I}_Q \right) \mathbf{I}_Q^\top.$

Note that an additional benefit of this simple modification is that it can greatly reduce computational cost. This is because nearest neighbour search and matrix inversion only ever need to be performed in dimension at most $q_0$, and for large $d$ we have found $q_0 \propto \sqrt{d}$ is more than adequate to achieve high accuracy.

## 3.2 A Full Model

Here we give a brief overview of our approach, which follows a standard format for the bagging framework. With settings for $k \in \mathbb{N}$, the number of neighbours; $q_0 \in [d]$, the number of randomly selected covariates used in a model; and $q \in [q_0]$, the dimension of each discriminant subspace, do the following for each $b \in [B]$:

1. Draw $\tilde{D}_b = \{(\mathbf{x}_{1^b}, y_{1^b}), ..., (\mathbf{x}_{n_B^b}, y_{n_B^b})\}$ without replacement from $D$.

2. Draw $Q_b$ of size $q_0$ without replacement from $[d]$.

3. For each $i \in [n_B]$ find $i_k^b$ and $i_k'^b$, the $k$-th in- and out-of-class neighbours of $\mathbf{I}_{Q_b}^\top \mathbf{x}_{i^b}$ from among $\{\mathbf{I}_{Q_b}^\top \mathbf{x}_{j^b}\}_{j \in [n_B], j \neq i}$.

4. Compute the discriminant matrix,

$$\hat{\Delta}_b = \mathbf{I}_{Q_b} \left( \mathbf{I}_{Q_b}^\top \frac{1}{n_B} \sum_{i=1}^{n_B} (\mathbf{x}_{i^b} - \mathbf{x}_{i_k^b})(\mathbf{x}_{i^b} - \mathbf{x}_{i_k^b})^\top \mathbf{I}_{Q_b} \right)^{-1} \left( \mathbf{I}_{Q_b}^\top \frac{1}{n_B} \sum_{i=1}^{n_B} (\mathbf{x}_{i^b} - \mathbf{x}_{i_k'^b})(\mathbf{x}_{i^b} - \mathbf{x}_{i_k'^b})^\top \mathbf{I}_{Q_b} \right) \mathbf{I}_{Q_b}^\top,$$

and let $\hat{\Delta}_b = \mathbf{U}_b \mathbf{\Lambda}_b \mathbf{U}_b^{-1}$ be its eigen-decomposition.

5. Form $g(\mathbf{x}|\tilde{D}_b)$ based on the class labels of the $k$ nearest neighbours of $\mathbf{U}_{b,1:q}^\top \mathbf{x}$ from $\{\mathbf{U}_{b,1:q}^\top \mathbf{x}_{1^b}, ..., \mathbf{U}_{b,1:q}^\top \mathbf{x}_{n_B^b}\}$, either by taking the proportions in each class or the indicator vector for the most frequent class. Here we have used $\mathbf{U}_{b,1:q} \in \mathbb{R}^{d \times q}$ to denote the first $q$ columns of $\mathbf{U}_b$.

Then obtain the final prediction by taking the mode of $\frac{1}{B} \sum_{b=1}^B g(\mathbf{x}|\tilde{D}_b)$.

### 3.2.1 Additional Useful Outputs

**Variable Importance** Interpretability of flexible predictive models is increasingly a point of focus, as modern methods typically rely on intricate relationships between the covariates and the response variable which may not be explicitly expressed within the model in any intelligible form. Variable importance scores are measures of the overall contribution of the covariates to the predictions made by a model. Although far from encapsulating the entirety of what a model has captured in the data these are nonetheless useful diagnostics for understanding which are the important variables driving the model's predictions.

The discriminant subspace framework offers an intuitive means by which the contribution of each variable to the model predictions may be quantified. In particular, if a variable lies within the discriminant subspace from one of the models in the ensemble, then it is natural to view that variable as important to the predictions from that particular model. On the other hand if a variable lies in the orthogonal complement of the subspace then this variable is unimportant. In general none of the variables will lie entirely within/without any of the discriminant spaces, but there will be some non-zero angles between them. If $\mathbf{U}$ has as columns a normalised (not necessarily orthogonal) basis for a subspace, then the cosines of the principal angles between the subspace and each of the variables in the cardinal basis can be seen as capturing the importance of each variable to the subspace and lie in the diagonal elements of $\mathbf{U}\mathbf{U}^\top$. To further capture the relative discriminatory information in the dimensions of the subspace, we weigh the basis vectors, which arise from the spectral decomposition of the discriminant matrix, by the eigen*values*. To determine the importance of the $j$-th variable to the entire ensemble we then average its importance values from each of the subspaces

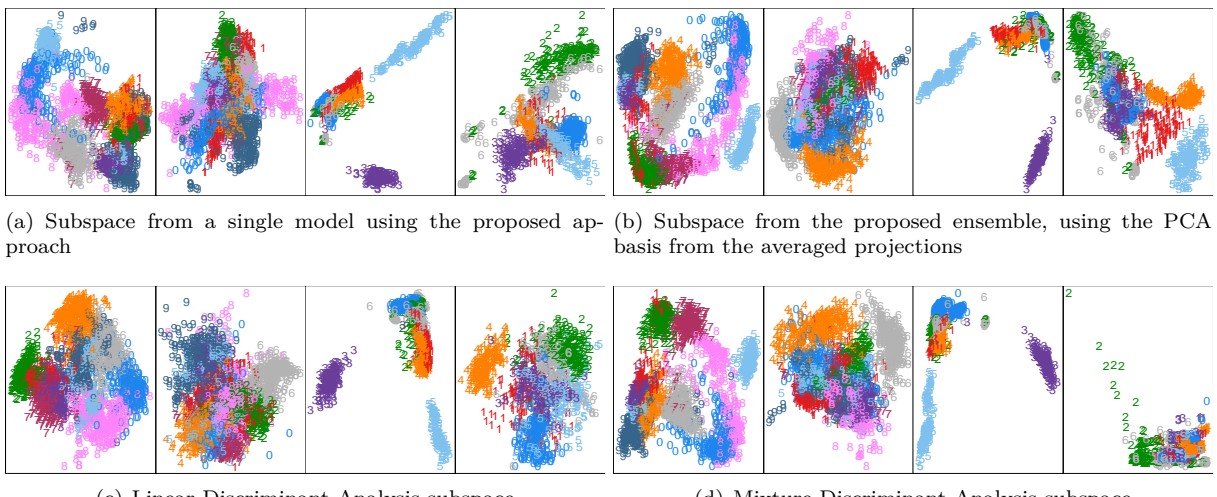

(a) Subspace from a single model using the proposed approach

(b) Subspace from the proposed ensemble, using the PCA basis from the averaged projections

(c) Linear Discriminant Analysis subspace

(d) Mixture Discriminant Analysis subspace

Figure 1: 2-Dimensional projections of the pen-based recognition of handwritten digits data set (left pair of plots in each subfigure) and the segmentation data set (right pairs).

in the model. Specifically, we take the importance of the $j$-th variable to the predictions from the ensemble to be $\frac{1}{B} \sum_{b=1}^{B} (\mathbf{V}_b \mathbf{V}_b^\top)_{jj}$, where $\mathbf{V}_b \in \mathbb{R}^{d \times q}$ has as columns the first $q$ columns of $\mathbf{U}_b \mathbf{\Lambda}_b^{1/2}$, where $\mathbf{U}_b$ and $\mathbf{\Lambda}_b$ contain the eigenvectors and eigenvalues of the $b$-th discriminant matrix, $\hat{\Delta}_b$.

**Visualisation** A further advantage of discriminant subspaces is the fact that they reduce the dimensionality of the observations. This can be beneficial for obtaining visualisations of the classes, and their separations from others, as well as the predictions made by a model. However, a single discriminant matrix within an ensemble, $\hat{\Delta}_b; b \in [B]$, is subject to fairly high variation. Moreover how to select from among multiple discriminant subspaces to obtain a single visualisation is not immediately obvious. We therefore aggregate the information from all discriminant subspaces, by determining the ensemble projection $\bar{\mathbf{P}} := \frac{1}{B} \sum_{b=1}^{B} \mathbf{U}_{b,1:q} \mathbf{U}_{b,1:q}^\top$ matrix. To obtain a visualisation of the observations in the aggregated discriminant subspace we project them onto the principal components basis vectors computed from the aggregate projected observations $\{\tilde{\mathbf{x}}_i\}_{i \in [n]}; \tilde{\mathbf{x}}_i := \bar{\mathbf{P}} \mathbf{x}_i$.

Figure 1 shows two examples, where projections of two of the data sets used in our experiments are shown. Each sub-figure contains four plots, with the left pair showing the first four discriminant projections of the sixteen dimensional pen-based recognition of handwritten digits data set (Alpaydin & Alimoglu, 1996), and the right pair those of the nineteen dimensional segmentation data set (ima, 1990). The points depict the projections of an independent test set separate from the "training" set used to obtain the actual projection directions, and the colours and point characters represent the individual classes. Figure 1(a) shows the result from a single discriminant subspace using all training observations and all variables to compute $\hat{\Sigma}_{in}^{-1} \hat{\Sigma}_{out}$, while Figure 1(b) shows the aggregated discriminant subspace from an ensemble of 100 models with $q_0 = \lfloor 0.75d \rfloor$ and $q = \lceil 0.5q_0 \rceil$. In both cases $k$ was set to three. Both show fairly clear separation of the majority of classes from others, with the ensemble showing these more clearly. For comparison we have also included the discriminant projections arising from LDA and MDA in Figure 1(c) and Figure 1(d), respectively. LDA shows good separation of the segmentation data set, but less so for the digits data set, while for MDA it is the reverse.

## 4 Experimental Results

### 4.1 Data Sets

For our experiments we considered all 162 classification data sets in the Penn Machine Learning Benchmarks database (Olson et al., 2017). We repeatedly sampled training and test sets from each data set, with training sets constituting 70% and test sets the remaining 30%. The number of times sampling training/test splits varied by overall sample size, $n$, as follows: $0 < n < 500 : 50$ times; $500 \leq n < 1000 : 20$ times; $1000 \leq n < 5000 : 10$ times; and $5000 \leq n : 5$ times. The one exception is that, due to the very large amount of compute time required for all experiments, training sets were capped at 7000 points and test sets at 3000 points (however, a different total 10 000 points was used in each training/test combination, where relevant).

Before passing the data sets to the different algorithms and models for fitting and prediction, all categorical variables were first one-hot-encoded. The only exception to this was in the case of the random forest models, since decision trees are able to handle categorical variables directly.

### 4.2 Classification Models and Tuning

Below we give a brief overview of the different models used for comparison, as well as how model selection was conducted for each. Although our primary interest is in the comparison between the proposed approach and the Random Forest models, we also include a number of alternatives for additional benchmarks and context.

1. BOPNN: The proposed approach (Bag Of Projected Nearest Neighbours), where each ensemble comprised a bag of 100 $k$NN models (i.e., $B = 100$). For each data set and training sample, thirty values for $k$ (the number of neighbours); $q_0$ (the size of the random subset of covariates sampled for each model); and $q$ (the number eigenvectors of each $\hat{\Delta}_b$ retained) were sampled uniformly as $k \sim U(\{1,...,5\})$; $q_0 \sim U(\{\lfloor p^{1/2} \rfloor,...,\min\{\lfloor 10p^{1/2} \rfloor, p\}\})$; and $q|q_0 \sim U(\{\lceil 0.5q_0 \rceil, q_0 - 1\})$ respectively. Models were fit for each setting of these hyperparameters and with the size of each bootstrap sample being 0.63 times the size of the training set ($\pi_B = 0.63$). The model with the highest Out-Of-Bag estimate for classification accuracy was then applied on the test set.

2. BOpNN: Equivalent to above, except no discriminant subspace was found for each model (or equivalently $q$ was always set to $q_0$). This variant is included primarily to give a clear indication of the benefit of including an adaptive learning step (the determination of a discriminant subspace) within a bagged model of otherwise lazy learners.

3. BNN$_\infty$: A bagged 1-NN model where the proportion of the sample included in each bootstrap sample is set equal to

$$\pi_B = \left(2\Gamma\left(2 + \frac{2}{p}\right)^2\right)^{\frac{p}{p+4}} \frac{1}{\hat{k}},$$

where $\hat{k}$ is an estimate of the optimal value for $k$ in a single $k$NN model, based on the leave-one-out cross-validation estimate for classification accuracy. This setting is a plug-in estimate for the asymptotically optimal value (Samworth, 2012). Note that for $\hat{k} = 1$ this proportion is greater than one, and in this case we simply set $\pi_B = 0.9$.

4. BNN: As above except where $\pi_B$ is tuned using OOB performance. This is included for comparison with BNN$_\infty$, to test the appropriateness of the plug-in estimate for the asymptotically optimal value for $\pi_B$.

5. $k$NN: A single $k$NN model with $k$ selected using the leave-one-out cross-validation estimate of classification accuracy.

6. ES$k$NN: The $k$NN ensemble which combines a selection from a large number of models fit to different bootstrap samples with different random subsets of the covariates (Gul et al., 2018). Unlike the bagged models above, this approach suffers unless the total number of $k$NN models is very large. Moreover, since no relevant OOB estimates for performance are available, the compute time required for this approach was

substantially greater than any of the other methods. As a result, we used a single 25% validation set for estimating performance and only selected $k$ from the set $\{1, ..., 10\}$. We fixed all other parameters equal to their defaults in the `ESKNN` package[2]. These settings deviated from the approach used by Gul et al. (2018) only in that a single validation set, as opposed to cross-validation, was performed to select $k$, and we used 501 initial models (as the default in the package) instead of 1001. Even with these changes, experiments with this approach required substantially larger compute time than any of the other methods. We also did not observe appreciably superior performance when using 1001 models instead of 501.

7. RF: The random forest classifier, as implemented in the `R` (R Core Team, 2024) package `randomForestSRC` (Ishwaran & Kogalur, 2019), available on CRAN. Following the same approach as for BOPNN, hyperparameter selection was conducted from 30 random selections based on OOB performance. The parameters which were tuned are "mtry" (the number of randomly selected covariates selected as candidates for each split in a tree), which was selected from the interval $[0.1p^{1/2}, \min\{p, 10p^{1/2}\}]$; and the minimum size of a leaf node in a tree, from $\{1, ..., 10\}$.

8. SVM: The Support Vector Machine classifier, where multiple classes were handled using the one-vs-one approach. We used the `LiquidSVM` implementation (Steinwart & Thomann, 2017), which uses a fast technique to approximate the kernel matrix but has nonetheless shown excellent performance in comparison with exact methods (Steinwart & Thomann, 2017). We used the default tuning grid and cross-validation settings provided in the implementation.

9. RDA: Regularised Discriminant Analysis (Friedman, 1989), where each class is modelled using a Gaussian distribution and class probabilities are determined using Bayes' rule. The means of the component distributions, $\hat{\boldsymbol{\mu}}_1, ..., \hat{\boldsymbol{\mu}}_K$, are determined by the sample means of the points from each class, while the covariance matrix of the $j$-th component is set equal

$$\hat{\Sigma}_j(\lambda, \gamma) := (1 - \gamma)\tilde{\Sigma}_j(\lambda) + \frac{\gamma}{p}\mathrm{trace}(\tilde{\Sigma}_j(\lambda))\mathbf{I},$$

$$\tilde{\Sigma}_j(\lambda) := \frac{1 - \lambda}{(1 - \lambda)n_j + \lambda n}\left(\sum_{i:y_i=j}(\mathbf{x}_i - \hat{\boldsymbol{\mu}}_j)(\mathbf{x}_i - \hat{\boldsymbol{\mu}}_j)^\top + \frac{\lambda}{1 - \lambda}\sum_{i=1}^{n}(\mathbf{x}_i - \hat{\boldsymbol{\mu}}_{y_i})(\mathbf{x}_i - \hat{\boldsymbol{\mu}}_{y_i})^\top\right),$$

where $\lambda$ and $\gamma$ are hyperparameters which must be chosen, and for which we used 5-fold cross-validation.

## 4.3 Summarising Classification Performance

Here we provide an overview of the classification performance of all methods across the collection of 162 data sets used for comparison. To combine the results from different data sets, which may have vastly different characteristics and represent classification tasks of varying difficulty, we first standardise the results. We consider two standardisations, and apply them to the classification accuracy values from the collection of models obtained on each data set, and each repetition of the sampling of training/test splits. For each model and each data set, we then take the two averages of its standardised accuracy values across the different test sets as its performance for that data set. Specifically, if $A_{m,i,t}$ is the accuracy of the $m$-th model on the $t$-th train/test split from the $i$-th data set, then we compute

$$A^*_{m,i,t} := \frac{A_{m,i,t} - \min_o A_{o,i,t}}{\max_o A_{o,i,t} - \min_o A_{o,i,t}}; \quad A^{**}_{m,i,t} := \frac{A_{m,i,t} - \overline{A_{\cdot,i,t}}}{s(A_{\cdot,i,t})},$$

where $\overline{A_{\cdot,i,t}}$ and $s(A_{\cdot,i,t})$ are the average and standard deviation of the accuracy values from all methods on the $t$-th train/test split of data set $i$. In the case of $A^*_{m,i,t}$ this simply maps the accuracy values to the interval $[0, 1]$, while $A^{**}_{m,i,t}$ is the common studentised value for $A_{m,i,t}$. The performance values for each method on a given data set are then just the averages of these standardised accuracies over $t$.

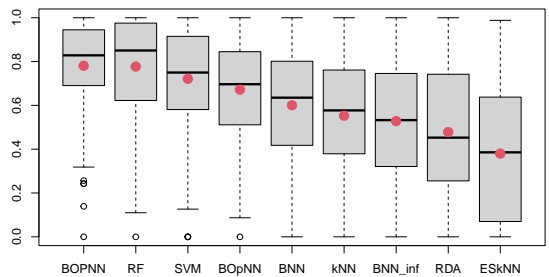 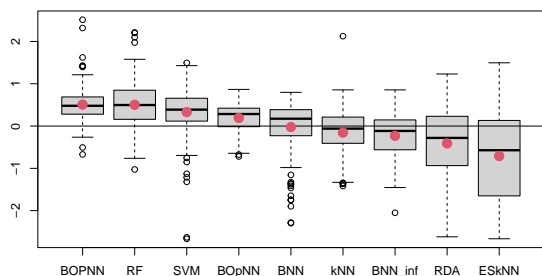

(a) Accuracy values mapped to the [0,1] interval

(b) Studentised accuracy values

Figure 2: Boxplots of accuracy distributions for different classification models, using two different standard-isations.

### 4.3.1 Accuracy Distributions

The distributions of standardised accuracy measures across all 162 data sets are shown in Figure 2. The main take-aways are summarised as

1. The average performances of BOPNN and RF are extremely similar, with BOPNN having slightly higher average but with RF having wider distributions. This is noteworthy since RF classifiers have commonly been referred to as excellent general purpose models; seldom substantially worse than any alternatives. These results suggest BOPNN similarly enjoys this feature, with arguably a better "worst-case" than RF due to a similar average and narrower distribution.

2. SVM outperforms all methods except the bagged models of adaptive non-parametric smoothers (BOPNN and RF).

3. BOpNN is substantially inferior to BOPNN, showing the importance of the adaptive learning step within the bagged model.

4. BOpNN is substantially superior to BNN. Although these two bagged models were tuned over disjoint collections of hyperparameters, the magnitude of the difference in performance is some indication that the purely randomised variable "selection" does indeed offer an improvement over its exclusion.

5. BNN is substantially superior to $BNN_\infty$. This suggests either that it is inappropriate to rely on the asymptotic theory for relatively small samples, or that the estimate for the asymptotically optimal $\pi_B$ itself is too unreliable.

6. ES$k$NN performed very poorly, and was the worst performing model on a large proportion of the data sets[3].

### 4.3.2 Pairwise Comparisons

Figure 3 shows the standardised accuracy values of all methods across all data sets. It is noteworthy that the dendrogram along the top axis of each sub-figure shows the performance of BOPNN is most similar to that of SVM. This is interesting since the average performances of BOPNN and RF are much more similar to one another than is either to the average from SVM. Yet, BOPNN seems to perform well/poorly on many of the same data sets as SVM, and to the extent that this overcomes the closer similarity of its average performance

---

[2]Previously on the Comprehensive R Archive Network (CRAN), and taken from `https://cran.r-project.org/src/contrib/Archive/ESKNN/`

[3]It is worth noting that the code released by the authors (previously on CRAN) included minor errors, such as populating arrays as though they were lists. Although we took care to correct these appropriately, it is not impossible that errors were made.

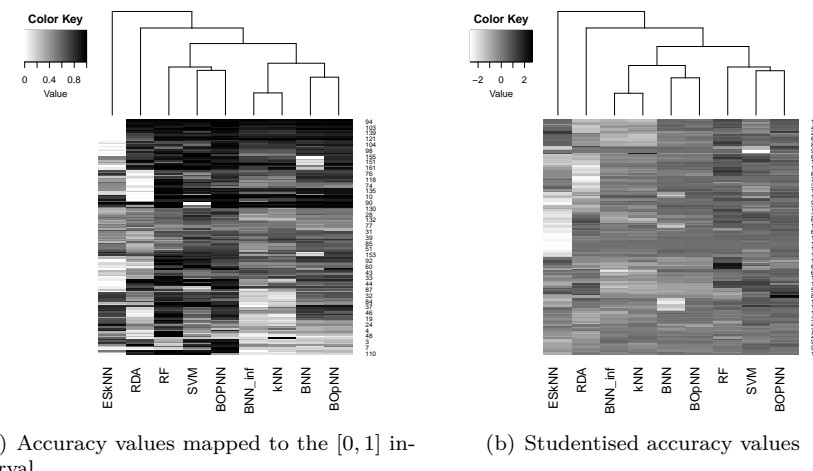

(a) Accuracy values mapped to the $[0, 1]$ interval

(b) Studentised accuracy values

Figure 3: Standardised classification accuracy across all data sets

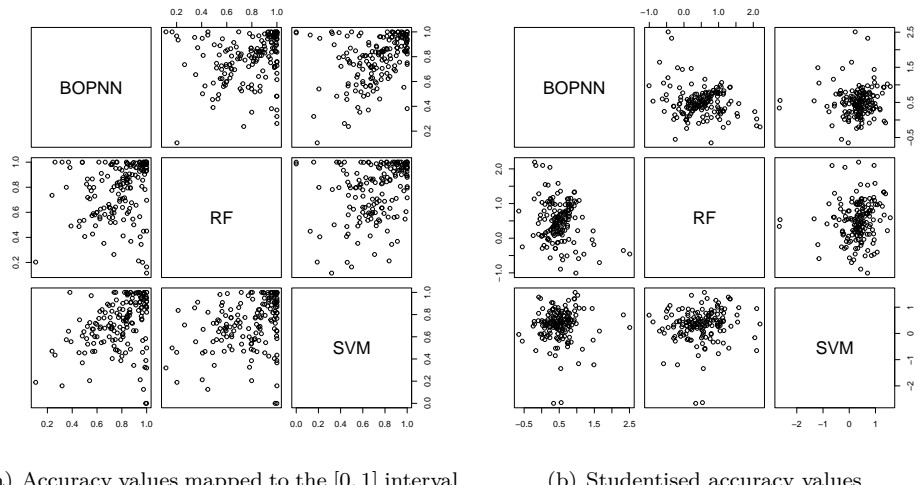

(a) Accuracy values mapped to the $[0, 1]$ interval

(b) Studentised accuracy values

Figure 4: Pairwise plots of standardised accuracy for the three top performing methods: BOPNN, RF and SVM

to that of RF. This can also be seen in Figure 4, where pairwise plots of the standardised accuracy values for these three models are shown. Interestingly the studentised performances of BOPNN and RF are *negatively* correlated whereas those of BOPNN and SVM are more or less uncorrelated.

In addition Table 1 shows the number of times, out of the total 162, the method listed row-wise achieved significantly superior performance to the method listed column-wise on a given data set. Significance was determined based on a paired Wilcoxon signed rank test (Wilcoxon, 1992) with test size 0.05[4]. Once again we are particularly interested in the comparison between BOPNN and Random Forests. Although the average performance of BOPNN is slightly superior, as shown in the previous subsection, we see here that RF outperformed BOPNN more frequently than the reverse. What is interesting to note is that RF both

---

[4]We acknowledge the arbitrariness of this test size, and do not mean to indicate any statistical relevance of these comparisons. Rather, we mean only to give a sense of the frequency with which each method outperforms each other method, while appropriately accounting for some of the randomness inherent in such a comparison. It is also worth pointing out that this is the smallest commonly used test size which is achievable on a sample of size 5, and we repeated the training/test splits on larger data sets five times.

|        | BOPNN | RF  | SVM | BOpNN | BNN | $k$NN | BNN$_\infty$ | RDA | ES$k$NN |
|-------:|-------|-----|-----|-------|-----|-------|--------------|-----|---------|
| BOPNN  | 0     | 37  | 53  | 60    | 70  | 99    | 107          | 115 | 107     |
| RF     | 46    | 0   | 57  | 71    | 84  | 101   | 101          | 108 | 114     |
| SVM    | 33    | 35  | 0   | 62    | 76  | 94    | 98           | 105 | 100     |
| BOpNN  | 6     | 19  | 33  | 0     | 23  | 73    | 76           | 92  | 92      |
| BNN    | 4     | 13  | 26  | 1     | 0   | 66    | 72           | 84  | 84      |
| $k$NN  | 6     | 19  | 21  | 7     | 25  | 0     | 16           | 72  | 81      |
| BNN$_\infty$ | 3 | 15  | 17  | 3     | 20  | 8     | 0            | 64  | 85      |
| RDA    | 9     | 18  | 15  | 29    | 41  | 44    | 45           | 0   | 64      |
| ES$k$NN | 8    | 11  | 17  | 21    | 31  | 41    | 45           | 55  | 0       |

Table 1: Pairwise comparative accuracy. Values in table indicate the number of times the method listed row-wise significantly outperformed the method listed column-wise. For example, BOPNN significantly outperformed RF 37 times, while RF significantly outperformed BOPNN 46 times. Significance was determined using a paired Wilcoxon signed rank test, with size 0.05.

outperforms the majority of the other methods more often than does BOPNN, and is also more frequently outperformed *by* them; indicated by greater values in the RF row *and* column than those of BOPNN.

### 4.4 Relationships between Performance and "Meta-data"

Here we investigate the relationships between the relative performance of the different methods, and the characteristics of the data sets. Each data set is characterised by six variables, $n$: the number of observations; $d$: the total number of variables after one-hot encoding; cat_prop: the proportion of binary variables in the one-hot encoded data; $K$: the number of classes; imbal: the class imbalance, defined as the variance of the class proportions; and compl: a measure of the complexity of the class decision boundaries, defined as $\log(\frac{A_{1NN}}{A_{NC}})$, where $A_{1NN}$ is the leave-one-out cross-validation estimate for the accuracy of the 1-nearest-neighbour classifier on the data, and $A_{NC}$ that of the nearest centroid classifier[5]. To capture dependence we computed the marginal correlations between the studentised accuracy of each method over all data sets and the data set characteristics (after log-transforming $n$ and $d$), as well as the Ordinary Least Squares (OLS) linear regression coefficients after standardising all variables[6]. These OLS coefficients give an indication of the correlations between the data set characteristics and the studentised performance *after accounting for the values of the other data set characteristics.*

Figure 5 shows heatmaps indicating the strengths of these relationships, with the marginal correlations in the left heatmap and the OLS coefficients in the right. Because these relationships are determined from a *standardised* accuracy measure, they give an indication of the relationships between the different methods and data set characteristics *relative to the other methods considered*. For example, we expect all methods will perform relatively better on larger data sets, all other things being equal. This would correspond with light colours in the first column of the right heatmap (positive OLS coefficients), however some methods are better/worse at leveraging larger samples than others and this is reflected by both positive AND negative OLS coefficients in the relationships with standardised accuracy. The lightest colours in this column indicate that RF, SVM and BOPNN might benefit more from larger samples than the other methods (with SVM benefiting the most). This is unsurprising as these are all very flexible models. Also unsurprisingly the parametric RDA does not leverage large samples well in comparison with the other models being considered. The "lazy" non-parametric $k$NN based models also do not leverage large samples as well as the adaptive models. ES$k$NN has an element of adaptiveness, in its selection from multiple randomised models, however the level of adaptiveness may not be sufficient to leverage larger samples particularly well. Some other noteworthy take-aways, for the purpose of this investigation, are:

1. BOPNN performs relatively very poorly in the context of class imbalance. This may be a result of the fact that the adaptive component of BOPNN comprises a linear transformation, and so applies globally, and

---

[5]The nearest centroid classifier simply classifies a point to the class whose mean vector is closest.

[6]we used studentised performance instead of the $[0, 1]$ mapped performance as their distributions are closer to Gaussian and, all other things being equal, may therefore be more appropriate when used in quantifying linear relationships. This is also the reason for log-transforming $n$ and $d$.

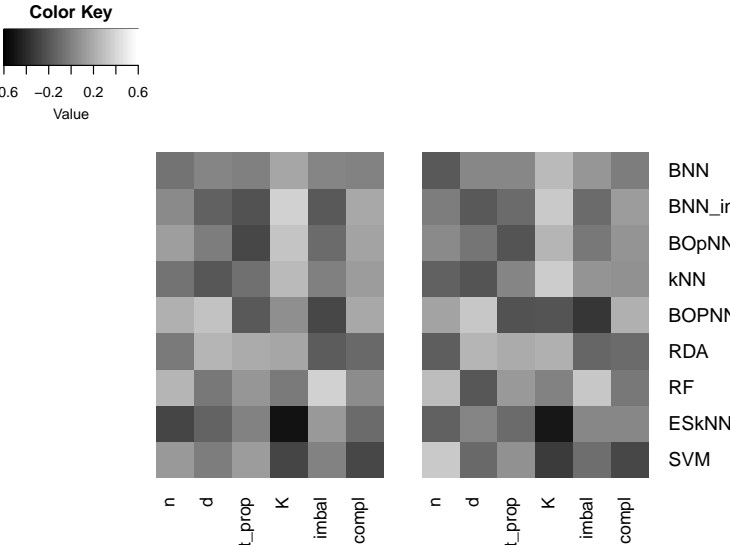

Figure 5: Relationships between studentised accuracy and data set characteristics. Left: marginal correlations, Right: standardised OLS coefficients

with the current implementation will be dominated by the larger classes. The local, more flexible component of BOPNN comes subsequently from the application of $k$NN on the transformed data.

2. BOPNN has a strong positive relationship with the "complexity" of the decision boundaries. However, it is possible that this result is somewhat artificial, given that the measure of complexity is governed by the performance of the 1NN model. Having said this, this performance is quantified relative to the other methods being compared, which includes numerous other nearest neighbour based methods.

3. The simplistic one-hot-encoding followed by Euclidean distance calculation currently employed in BOPNN may be inappropriate, as indicated by its comparatively poor performance when a large number of categorical variables are present.

4. Although BOPNN performed reasonably well on data sets with a large number of classes (light colour in the left heatmap for BOPNN and $K$), this appears to be almost fully determined by the other characteristics of those data sets with large $K$ since it has a strong *negative* OLS coefficient for $K$.

5. Compared with Random Forests, BOPNN may be better suited to higher dimensional examples but may leverage larger samples less well.

### 4.4.1 "Meta-data" Distributions

For completeness here we also provide illustrations of the (marginal) distributions of the data set characteristics. Figure 6 shows all the values of all six data set characteristics across the 162 data sets. In addition, the performance of BOPNN is indicated via the sizes of the points, with larger points aligning with instances where BOPNN performed well relative to the other methods considered.

## 5   Concusions

In this paper we discussed the importance of including an adaptive learning step within the context of bootstrap aggregating, or "bagging", and proposed a simple adaptive $k$NN model for use within "bagged" ensembles. The adaptive step is achieved by finding discriminant subspaces which enhance the class separation, as captured by $k$NN classifiers. The discriminant subspace framework naturally leads to measures of

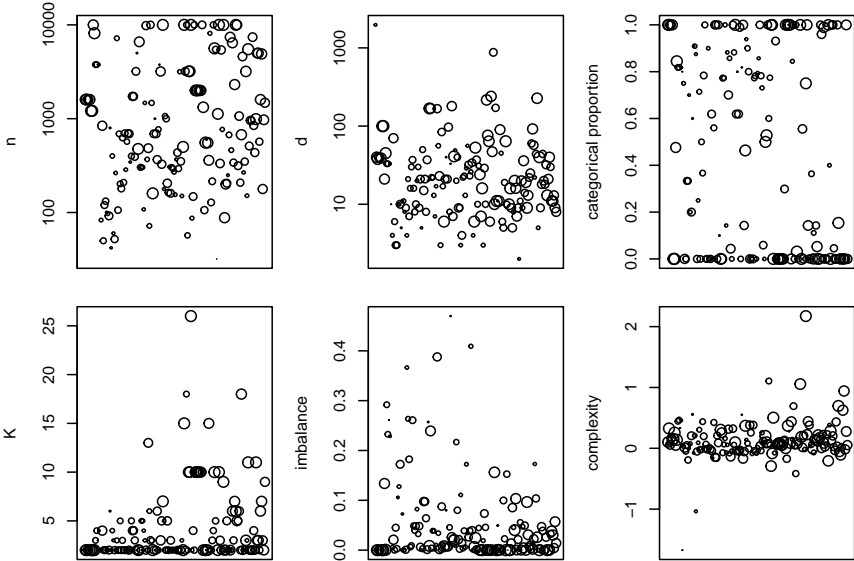

Figure 6: Distributions of "meta-data", with $n$ and $d$ on a log scale. The sizes of points are related to performance of BOPNN with larger points corresponding with better BOPNN performance, relative to the other methods considered.

variable importance and offers instructive visualisations of the classes through projections into the discriminant subspaces (or an aggregated variant incorporating the entire ensemble).

In an extensive set of experiments we documented the strong potential offered by the proposed approach. Noteworthy findings are that across varied contexts the proposed approach is more or less on par with Random Forests, on average, but that the particular data sets on which the proposed approach may be more or less suited in fact align better with Support Vector Machines. Potential directions for improvement of BOPNN include an alternative, but computationally efficient, way to incorporate categorical variables, as well as strategies to enhance performance with highly imbalanced class proportions.

## Acknowledgements

We would like to thank the reviewers for their valuable comments which helped to improve the quality of this work.

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

# A    Additional Experimental Results

## A.1    Alternative "Discriminant Subspaces"

As mentioned in Section 3.1 it is possible to pair $k$NN classifiers with other dimensionality reduction techniques/discriminant subspaces within a bagged ensemble. However, as we discussed, the appropriateness of different subspaces will depend on the type of classifier being used, and the proposed approach is motivated specifically by minimising the in-class and maximising the out-of-class near neighbour distances and so we anticipate that this will pair particularly well with $k$NN. It is also the case that the discriminant matrices used within the proposed approach use much finer detail in the data than, e.g. the LDA discriminant matrix, and so we expect greater differentiation in the individual models across bootstrap samples; a key factor in leveraging the benefits of the model averaging framework.

Here we present experiments documenting the improved performance using the proposed approach, and bagged ensembles of $k$NN models fit within LDA discriminant subspaces and Principal Components (PCA) subspaces. We include PCA for illustrative purposes, as it represents a fundamental *unsupervised* dimension reduction model, and so it represents an alternative which is not completely randomised but is not adaptive to the relationships between the response and covariates. In addition, since there are only at most $K-1$ non-arbitrary dimensions in the LDA subspace, when using more than $K-1$ discriminant dimensions we combined the LDA subspace with the leading PCA dimensions within their orthogonal complement. For as direct a comparison as possible we used exactly the same tuning strategy for these variants as for BOPNN.

The distributions of standardised accuracy measures are shown in Figure 7. We only show the better performing and most relevant models from the point of view of these comparisons, however the standardisations were determined using the entire collection of (now 11) methods. Of note is that (i) using the LDA subspace (BOLDANN) leads to performance similar to that of SVM, on average; (ii) applying PCA appears to be detrimental to performance, in general, as the performance of BOPCANN is inferior to that of BOpNN; and (iii) after including these two additional models the average studentised performance of RF is now marginally above that of BOPNN, however the the other points of comparison are unchanged, and also the average performances of RF and BOPNN remain extremely similar.

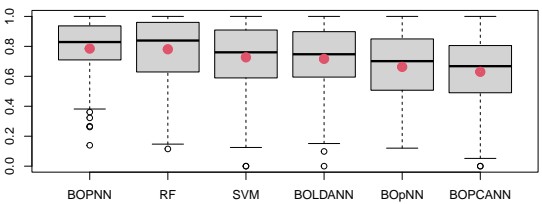
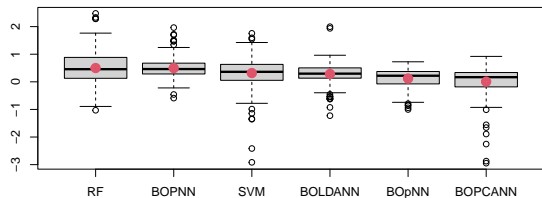

(a) Accuracy values mapped to the $[0, 1]$ interval

(b) Studentised accuracy values

Figure 7: Standardised accuracy distributions relevant for comparison with the alternative bagged $k$NN models applied on "discriminant subspaces".

## A.2    Default Settings and the Effect of Varying $k$

In preliminary experiments with BOPNN we worked with default settings of $q_0 = \left\lceil \min\{0.75d, 5\sqrt{d}\} \right\rceil$; $q = \left\lceil 0.5q_0 \right\rceil$ and $k = 3$, and found these to be fairly reliable. However, unlike with RFs where the dimensionality restriction through randomised variable selection/restriction only applies locally (i.e. separately at each node in the trees), the expressiveness of each of the models in BOPNN is fundamentally limited by the value of $q$ (and hence $q_0$ as well). In particular when the number of classes, $K$, is reasonably large then it may be impossible to separate all classes accurately in even the theoretically optimal subspace if its dimension is

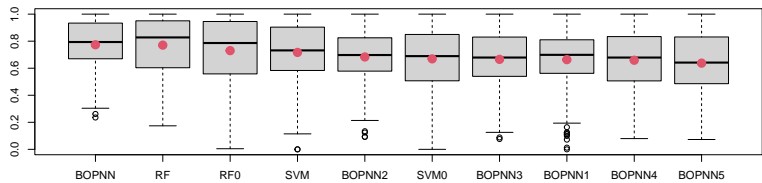

(a) Accuracy values mapped to the $[0, 1]$ interval

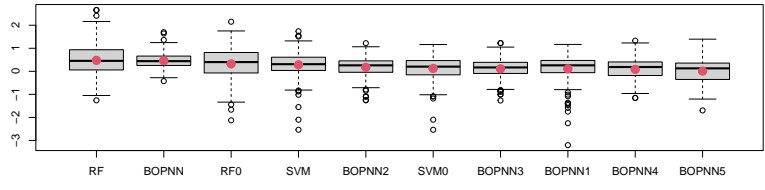

(b) Studentised accuracy values

Figure 8: Standardised accuracy distributions relevant for comparison with default settings.

too low. It is likely, therefore, that settings of $q_0$ and $q$ should also depend on $K$. It may also simply be that this limitation in expressiveness is unavoidable in any general sense, and that some degree of tuning is necessary to consistently yield very good performance.

Here we present results analogous to those in Section A.1, except that we have included default variants of RF and SVM (named RF0 and SVM0 below) as well as BOPNN models with the above settings of $q_0$ and $q$ and for $k = 1, 2, 3, 4$ and $5$. The distributions of standardised accuracy values are shown in Figure 8. As before the standardisations have been determined using all (now 16 models; 9 main models plus two defaults from competing methods and 5 fixed settings for BOPNN) methods, but we only show the most relevant ones here.

Noteworthy observations are that (i) none of the fixed setting BOPNN models is competitive with the tuned RF, SVM and BOPNN models; (ii) default settings for RF leads to similar overall performance to that of *tuned* SVM; (iii) the fixed setting of $k = 2$ provided the best overall performance of BOPNN (however whether this is generalisable to potentially improved defaults for $q_0$ and $q$ is unclear); (iv) the default variant of SVM and the best fixed BOPNN model (BOPNN2) have very similar overall performance; and (v) although not directly apparent from these figures RF0, SVM0 and BOPNN1-4 all outperform all of the other (tuned) models not included in these plots and only BOpNN outperforms BOPNN5 from among them.

One other point is that, as we saw in Section A.1, upon including additional models the average studentised performance of RF is greater than that of BOPNN (in their tuned variants). However, it is worth pointing out that studentisation in the context where there is a high degree of correlation will tend to "over standardise" the performance of groups of models whose performance is highly correlated. The reason for this is that on data sets where the performance of these models is either well above or below the others, this over/under "achievement" is normalised by the presence of multiple models with similar extreme performance. In particular, if we only include one default setting of BOPNN (any of the five) in the standardisations, then the average studentised performance of BOPNN remains above that of RF.

## A.3  Computational Complexity and Running Times

The worst case complexity of BOPNN is $\mathcal{O}(B(kn^2 q_0 + q_0^3))$, since the main computational steps within each of the $B$ models in the ensemble involve finding the $k$ in- and out-of-class neighbours in dimension $q_0$, and for

|       |       |      | RF | | BOPNN |
| $n$ | $d_0$ | $d$ | Wall-clock | CPU | Wall-clock |
|------|------|------|------|------|------|
| 1600 | 1000 | 1969 | 3.8 | 19.4 | 121.9 |
| 7000 | 16 | 16 | 3.6 | 16.8 | 10.5 |
| 7000 | 784 | 864 | 14.6 | 81.0 | 102.2 |

Table 2: Average running times of RF and BOPNN on three of the larger data sets used in our experiments.

inversion and eigen-decomposition of $q_0 \times q_0$ matrices. However, when $q_0$ is relatively small the running time of nearest neighbour search is closer to $n \log(n)$ than to $n^2$, meaning that for cases of moderate dimensionality the running time is close to $\mathcal{O}(B(kn \log(n)))$. A potential avenue for improvement computationally is to include a variable screening stage, to obtain heuristic variable importance scores for non-uniform sampling of covariates so that $q_0$ can be set relatively small even when the total dimension is large. It is worth pointing out that in order for the OOB estimates of performance to be valid, these variable importance scores would need to be computed *within each* model in the ensemble.

To illustrate the poor scaling with dimension, we here report the empirical running times of BOPNN and RF (with their default settings) on three of the larger data sets used in the experiments; one with very large $d$, one with fairly large $n$ and small $d$, and one with fairly large $n$ and $d$. We only consider the larger data sets as these are the most critical when it comes to running time comparisons. It is worth pointing out that our implementation of BOPNN has not been optimised for speed, and runs in serial on a single thread. On the other hand the RF implementation in the package `randomForestSRC` is highly optimised and parallelised. As a result we report both the wall-clock time of RF and the total CPU time, whereas for BOPNN these are of course very similar and we only report wall-clock time.

The average running times from ten runs on each data set are shown in Table 2. Here we have used $d_0$ to be the number of raw covariates and $d$ to be the number of covariates after one-hot-encoding. We include both since RF handles categorical variables directly, and so operates in dimension $d_0$. Aside from the very high dimensional example the total CPU time required for fitting RF models and BOPNN models is similar, however the multithreading used in the implementation of RFs in `randomForestSRC` means the actual running times are not currently comparable. Nonetheless these experiments indicate that BOPNN has the potential to be computationally competitive with efficient RF implementations, potentially with the exception of when the number of covariates is very large.

