. ~~The classification task is then to use this~~ This *training* sample, $D$, ~~to obtain a model~~ is then used to fit a model, $g(\cdot|D)$ ~~which, given a~~, which is used to predict the class label for any given *query point*, $\mathbf{x} \in \mathbb{R}^d$~~, is able to provide a prediction for the class to which $\mathbf{x}$ belongs~~.

Bagging operates by resampling from $D$ multiple times to produce $B$ *bootstrap* samples, $\tilde{D}_1, ..., \tilde{D}_B$, and then combining the resulting models, $g(\cdot|\tilde{D}_b); b = 1, ..., B$, to obtain a final predictive model. Note that whether each bootstrap sample is obtained by sampling with or without replacement often has relatively little impact on the performance of the overall model~~, and for~~. For ease of exposition we ~~consider~~ assume sampling without replacement, and in such a way that each bootstrap sample contains ~~$n_B < n$ observations~~ $n_B = \lceil \pi_B n \rceil$ observations, where $\pi_B \in (0, 1)$.

### 2.1 Bagging and Variance Reduction

~~Although requiring considerably greater computational investment than fitting a single predictive model, bagging~~ Bagging can be remarkably effective in reducing the variance of flexible predictive models. This is most conveniently communicated when combining the individual models through averaging, i.e., when using

$$g^{(bag)}(\mathbf{x}|D) = \frac{1}{B} \sum_{b=1}^{B} g(\mathbf{x}|\tilde{D}_b). \tag{1}$$

~~Although averaging classification models whose outputs are themselves class labels is clearly nonsensical, the averaging formulation in Eq. (1) is applicable in this context if we simply consider that each model either outputs an indicator vector for the class allocation of an input vector, $\mathbf{x}$, or an estimate for the conditional distribution of $Y|X = \mathbf{x}$. In the latter case the quantity $g^{(bag)}(\mathbf{x}|D)$ may also be seen as an estimate for the distribution of $Y|X = \mathbf{x}$, whereas in the former $g^{(bag)}(\mathbf{x}|D)$ is better interpreted as an estimate for the~~

~~distribution of $g(\mathbf{x}|\tilde{D}_0)$, where $\tilde{D}_0$ is a sample of size $n_B$ drawn directly from the underlying population. In either case a final class prediction can be obtained by taking the mode of $g^{(bag)}(\mathbf{x}|D)$.~~

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

 ~~, is the adaptive determination of the regions of influence of each point. This benefit manifests both through how it allows DTs~~ are at least two-fold: (i) it enables the models to exploit the relationships between the covariates and the response~~, and also through the fact that the~~; and (ii) it allows the region of influence of a point ~~is dependent~~ to depend on the entire sample, and not only on points which are very local to it. This latter fact induces further differentiation across the outputs of ~~DTs~~ models fit to different bootstrap samples, and so reduces their correlation.

Making $k$NN classifiers adaptive by optimising the distance metric used in the nearest neighbour search is intuitively pleasing, and in principle has the potential to achieve the same benefits as those described above for DTs. ~~But where these~~ However, most adaptive $k$NN methods are ~~limited is their computation time, which pales in comparison with the speed of DTs, and all but precludes their use within bagged ensembles. A fact which~~

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

~~. However, *we have found* ~~the formulation described above to yield better performance within bagged ensembles. Intuitively the leading eigenvectors of $\hat{\Sigma}_{in}^{-1}(\hat{\Sigma} - \hat{\Sigma}_{in})$ result in projections on which the in-class neighbours are pushed nearer to one another,~~ *substantially superior performance using the eigenvectors of $\hat{\Sigma}_{in}^{-1}\hat{\Sigma}_{out}$. We believe this is at least partly due to the* ~~term $\hat{\Sigma}_{in}^{-1}$, while the term $(\hat{\Sigma} - \hat{\Sigma}_{in})$ ensures this doesn't arise simply by pushing allpoints closer to one another. On the other hand the term $\hat{\Sigma}_{out}$ in our approach explicitly contributes to class discrimination by pushing points away from their out-of-class neighbours. Note that the matrix~~ *fact that $\hat{\Sigma}_{in}^{-1}\hat{\Sigma}_{out}$* ~~also~~ *uses a greater amount of information from the sample than does $\hat{\Sigma}_{in}^{-1}(\hat{\Sigma} - \hat{\Sigma}_{in})$, and therefore intuitively leads to greater diversity across bootstrap samples.*

### 3.2 ~~Other Practicalities~~

~~Here we comment on some further practical aspects of the proposed approach of bagging the predictions arising from~~

**Remark 5** *It is worth noting that we use only the k-th in- and out-of-class neighbours of each point when determining the discriminant subspace despite the fact that the classification of a point after its projection is based on all of its k nearest neighbours. This is ultimately because the density, as quantified by kNN* ~~models obtained on different discriminant subspaces. For each $b \in [B]$, let $\hat{\Delta}_b$ be the discriminant matrix whose leading eigenvectors, in the columns of the matrix $\mathbf{U}_b$, determine the discriminant subspace for the b,~~ *is based only on the k-th* ~~model in the ensemble. $\hat{\Delta}_b$ represents the matrix $\hat{\Sigma}_{in}^{-1}\hat{\Sigma}_{out}$ computed from the b~~*neighbour distance. Perhaps more intuitively, minimising the k-th* ~~bootstrap sample, but we introduce this new notation to accommodate a slight variation, described below.~~*in-class neighbour distance has the effect of making the ball containing all k nearest neighbours as small as possible.*

### 3.1.1 Additional Diversity Through ~~Randomisation~~Randomised Variable Selection

As described previously, additional randomness across different models (e.g. through randomised variable selection) can be beneficial if the resulting decrease in accuracy of the models arising from this randomness (which may also be seen as a reduction in information) is outweighed by the effect of additional diversification of the individual models. Adaptive models mitigate the reduction in accuracy arising from reduced information, when compared with lazy learners, as they better exploit what information *is* available and are better suited to filtering out noise.

~~We have found that within our proposed approach~~ In our preliminary experiments it became clear that restricting the discriminant subspaces each to lie within their own randomly determined higher dimensional subspace does indeed improve performance in general. How we ~~implement this is to, for each $b \in [B]$, randomly sample~~ achieve this is simply by only using information from a (random) subset of covariates in each model in an ensemble. Specifically, let $Q \subset [d]$ be a subset of the ~~covariates, $Q_b \in [d]$, of size $q_0 \le d$, and then compute $\hat{\Sigma}_{in,b}$~~ total covariates (which in practice is determined randomly), with $|Q| = q_0$, and ~~$\hat{\Sigma}_{out,b}$ from the b-th bootstrap sample but restricted to the subspace containing the variables in $Q_b$. The discriminant matrix $\hat{\Delta}_b$ is then simply equal to $\mathbf{I}_{Q_b}\hat{\Sigma}_{in,b}^{-1}\hat{\Sigma}_{out,b}\mathbf{I}_{Q_b}^{\top}$ where $\mathbf{I}_{Q_b} \in \mathbb{R}^{d \times q_0}$ has~~ let $\mathbf{I}_Q \in \mathbb{R}^{d \times q_0}$ have as columns the cardinal basis vectors for the ~~variables in $Q_b$.~~ dimensions in $Q$. Then define $i_k$ and $i_k'$ as before, except with distances only computed with the variables in $Q$. We then define the *discriminant matrix*, whose eigenvectors define the discriminant subspace, simply by $\hat{\Delta} := \mathbf{I}_Q \left( \mathbf{I}_Q^{\top}\hat{\Sigma}_{in}\mathbf{I}_Q \right)^{-1} \left( \mathbf{I}_Q^{\top}\hat{\Sigma}_{out}\mathbf{I}_Q \right) \mathbf{I}_Q^{\top}$.

~~This randomised reduction in dimensionality is beneficial both from the point of view of classification accuracy of the overall ensemble, and importantly also substantially reduces the computational complexity of the method~~Note that an additional benefit of this simple modification is that it can greatly reduce computational cost. This is because nearest neighbour search and matrix inversion only ~~needs~~ ever need to be performed in dimension at most $q_0$, and for large $d$ we have found $q_0 \propto \sqrt{d}$ is more than adequate to achieve high accuracy.

### 3.1.2 ~~Variable Importance~~

## 3.2 A Full Model

Here we give a brief overview of our approach, which follows a standard format for the bagging framework. With settings for $k \in \mathbb{N}$, the number of neighbours; $q_0 \in [d]$, the number of randomly selected covariates used in a model; and $q \in [q_0]$, the dimension of each discriminant subspace, do the following for each $b \in [B]$:

1. Draw $\tilde{D}_b = \{(\mathbf{x}_{1^b}, y_{1^b}), ..., (\mathbf{x}_{n_B^b}, y_{n_B^b})\}$ without replacement from $D$.

2. Draw $Q_b$ of size $q_0$ without replacement from $[d]$.

3. For each $i \in [n_B]$ find $i_k^b$ and $i_k'^b$, the $k$-th in- and out-of-class neighbours of $\mathbf{I}_{Q_b}^\top \mathbf{x}_{i^b}$ from among $\{\mathbf{I}_{Q_b}^\top \mathbf{x}_{j^b}\}_{j \in [n_B], j \neq i}$.

4. Compute the discriminant matrix,

$$\hat{\Delta}_b = \mathbf{I}_{Q_b} \left( \mathbf{I}_{Q_b}^\top \frac{1}{n_B} \sum_{i=1}^{n_B} (\mathbf{x}_{i^b} - \mathbf{x}_{i_k^b})(\mathbf{x}_{i^b} - \mathbf{x}_{i_k^b})^\top \mathbf{I}_{Q_b} \right)^{-1} \left( \mathbf{I}_{Q_b}^\top \frac{1}{n_B} \sum_{i=1}^{n_B} (\mathbf{x}_{i^b} - \mathbf{x}_{i_k'^b})(\mathbf{x}_{i^b} - \mathbf{x}_{i_k'^b})^\top \mathbf{I}_{Q_b} \right) \mathbf{I}_{Q_b}^\top,$$

   and let $\hat{\Delta}_b = \mathbf{U}_b \mathbf{\Lambda}_b \mathbf{U}_b^{-1}$ be its eigen-decomposition.

5. Form $g(\mathbf{x}|\tilde{D}_b)$ based on the class labels of the $k$ nearest neighbours of $\mathbf{U}_{b,1:q}^\top \mathbf{x}$ from $\{\mathbf{U}_{b,1:q}^\top \mathbf{x}_{1^b}, ..., \mathbf{U}_{b,1:q}^\top \mathbf{x}_{n_B^b}\}$, either by taking the proportions in each class or the indicator vector for the most frequent class. Here we have used $\mathbf{U}_{b,1:q} \in \mathbb{R}^{d \times q}$ to denote the first $q$ columns of $\mathbf{U}_b$.

Then obtain the final prediction by taking the mode of $\frac{1}{B} \sum_{b=1}^{B} g(\mathbf{x}|\tilde{D}_b)$.

### 3.2.1 Additional Useful Outputs

**Variable Importance**  Interpretability of flexible predictive models is increasingly a point of focus, as modern methods typically rely on intricate relationships between the covariates and the response variable which may not be explicitly expressed within the model in any intelligible form. Variable importance scores are measures of the overall contribution of the covariates to the predictions made by a model. Although far from encapsulating the entirety of what a model has captured in the data these are nonetheless useful diagnostics for understanding which are the important variables driving the model's predictions.

The discriminant subspace framework offers an intuitive means by which the contribution of each variable to the model predictions may be quantified. In particular, if a variable lies within the discriminant subspace from one of the models in the ensemble, then it is natural to view that variable as important to the predictions from that particular model. On the other hand if a variable lies in the orthogonal complement of the subspace then this variable is unimportant. In general none of the variables will lie entirely within/without any of the discriminant spaces, but there will be some non-zero angles between them. If $\mathbf{U}$ has as columns a normalised (not necessarily orthogonal) basis for a subspace, then the cosines of the principal angles between the subspace and each of the variables in the cardinal basis can be seen as capturing the importance of each variable to the subspace and lie in the diagonal elements of $\mathbf{U}\mathbf{U}^\top$. To further capture the relative discriminatory information in the dimensions of the subspace, we weigh the basis vectors, which arise from the spectral decomposition of the discriminant matrix, by the eigen*values*. To determine the importance of the $j$-th variable to the entire ensemble we then average its importance values from each of the subspaces in the model. Specifically, ~~if $\hat{\Delta}_b = \mathbf{U}_b \mathbf{\Lambda}_b \mathbf{U}_b^{-1}$ is the standard spectral decomposition of the $b$-th discriminant matrix, with eigenvalues in the diagonal matrix $\mathbf{\Lambda}_b$ listed in decreasing order, then the~~