# OpenReview forum: "Bags of Projected Nearest Neighbours: Competitors to Random Forests?"
_TMLR — Accepted by TMLR_

### Review · Reviewer_yi1y · 2025-07-29

**Summary Of Contributions:**

In their paper "Bags of projected nearest neighbours: competitors to random forests?", the authors suggest a new approach to bagging kNN classifiers. The authors argue that in bagging it is important that models are sufficiently adaptive to the relationship between X and y, and kNN classifier does not satisfy this property, which is why bagged kNN classifiers perform poorly. To remedy that, the authors suggest to use a modified kNN classifier, where neighbors are identified in a low-dimensional subspace identified in a supervised way similarly to LDA. They conductive a massive benchmark using 100+ datasets and show that this approach performs similarly to random forests and outperforms other bagging+kNN strategies.

Strengths: I enjoyed reading the paper. The introduction is very clear, the experiments are comprehensive, the authors do reasonable ablation experiments. The results are strong. Good work.

Disclaimer: I was not familiar with any papers in the related work section or with any other work on bagging kNN classifiers.

Weaknesses: I suggest some clarifications and some additional visualizations/analysis.

Overall, this can definitely be published in TMLR after a minor revision.

**Audience:**

Yes

**Audience Explanation:**

Interesting topic relevant for machine learning community.

**Broader Impact Concerns:**

None.

**Claims And Evidence:**

Yes

**Claims Explanation:**

Claims are supported by clear comprehensive experiments.

**Requested Changes:**

MAJOR COMMENTS

* Section 3.1: I am not sure how the classifier is working exactly, I feel something is missing from this section. The section explains how to construct a low-dimensional subspace based on parameter $k$. What happens then? You project the entire dataset onto this subspace and perform kNN classification inside that subspace? I.e. the kNN neighbors are identified in the subspace? Is the subspace constructed using only the training data, and both train+test are then projected there? Please spell all of this out in Section 3.1.

* Subspace construction depends on $k$ as a parameter. Then later you are going to use kNN classifier which also has parameter $k$. Why is the same value of $k$ used in these two contexts? Also, kNN classifier uses all neighbors from 1st to kth. But in equation (3) you only use the kth neighbors. Why?

* Could one instead of Equation (3) simply take standard LDA within- and between- class covariances? Without any notion of k? So first project the data onto LDA subspace, then perform kNN classification? Would this be a reasonable strategy for bagging? If not, could you discuss why not?

* Section 4.1: please add some information about the collection of 162 datasets. What is the distribution of sample sizes (min / mean / max), feature dimensionality, number of classes, etc. Maybe add it as a table.

* Compute time: first, please report the total compute time required for all your experiments. Second, discuss runtime of BOPNN vs RF.

* I would suggest to move Appendix A into section 4.2. Currently section 4.2 is unclear without looking into Appendix A, and you don't have space limits in TMLR.

* Section 4.2: BNN is unclear. Also it was not described in Related Work section, why not?

* On the other hand, Related Work described Cannings & Samworth 2017 but you are not comparing against it, why not?

* It could be nice to add scatter plots of accuracy achieved e.g. by BOPNN vs RF across datasets (so 162 points on the scatter plot). Also maybe of BOPNN vs BOpNN or kNN -- that would show that BOPNN is superior. It's just a different way of visualizing these comparisons.

* You used a search over 30 param combinations for BOPNN. Do you have a sense for which are sensible defaults? E.g. k=5, qo = sqrt(p), q = q0/2 or something. Can you discuss that? Can you report the performance of hyperparam-free BOPNN where all these hyperparams are fixed to sensible defaults?


MINOR COMMENTS

* I would reformulate the title into a statement: "Bags of projected nearest neighbours are competitive with random forests"

* The phrase "computationally efficient to compute" appears at least twice and should be reformulated.

* Section 2.2: I would appreciate a bit more detail about Cannings & Samworth 2017 and Gul et al 2018. Can you explain in more detail how they work? These two papers seem to be most relevant for your work.

* Section 4.1: "number of repetitions": clarify what is meant by "repetition", otherwise readers may be confused and think you are talking about bootstrap repetitions.

* Section 4.3.2: Wilcoxon signed rank test: unclear what exactly was tested. For each given dataset and a pair of algorithms what were you testing exactly?

* Appendix A: kNN classifier, what was the grid for k values?

* Appendix A: I did not understand BNN_inf or BNN. More details needed.

---

> ### Author Response · Authors · 2025-08-04
> **Some initial responses and clarifications**
>
> Dear reviewer,
>
> first off, thank you very much for your time and effort (especially such a remarkably quick turnaround!).
>
> Before making adjustments and/or conducting more experiments, I hope you don't mind if we first respond and clarify on a few points:
> - Section 3.1: Thanks a lot for this comment. It is obviously very important the actual methodology is clear. From your comments it seems as though your understanding is correct, but nonetheless your not feeling certain as a reader is very telling of how it was presented. We will revise.
> - Value of $k$ across subspace determination and classifiers: We do indeed use the same value for $k$ in both, and this is largely because tuning over multiple such values adds (as far as we feel, unnecessarily) to the overall computation. Your comment about the dependence of the subspace on only the $k$-th neighbour (unlike the classifier) is a good one. Ultimately if we shift the points closer to their $k$-th neighbours the method is trying to "compress" the ball containing all $k$ nearest neighbours, and in fact the difference between using only the $k$-th and all up to the $k$-th, in terms of overall quality of the solutions, is very minimal. If you believe the method would be more "elegant", we would be happy to make the change. It would just mean running the experiments again, although this may be needed anyway to address some other points.
> - LDA subspace instead?: In principle, absolutely one could. As we describe in the paper, we actually benefit from more variation across subspaces and the LDA variants are much more "stable" in this regard. We may also miss subtler structure than just separation by means (relative to (co)-variance) which the NN classifiers can use (unlike the Gaussian LDA classifiers). It is also important that the LDA subspace is only non-arbitrary up to K-1 dimensions (for K classes). We can elaborate in the paper
> - Distribution of data set "metadata": Absolutely, we can add this. In relation to this, and some other points, as far as we are aware there is a page limit of 12, which is why we moved the details of the classifiers and how we did hyperparameter tuning to the appendix. There is a 16 page "variant" but this requires justification for why we need a "long paper" format.
> - Compute time: We did not record overall compute time, but as mentioned may need to re-run the experiments, or at least some of these. As for the comparison between BOPNN and RF: we are not professional programmers, whereas the faster RF implementations are hyper-efficient. Actual absolute run-times for our implementation of BOPNN may not be that relevant. We are happy to include them if you think they are instructive. Perhaps better would be something about computational complexity, which off the top of my head is probably $\mathcal{O}(B(n log(n) q_0 + q_0^3))$, where $B$ is the number of bags and $q_0$ is the dimension of the sampled subspace. For RF I imagine this is something $\mathcal{O}(B(n \times mtry \times d))$ where $d$ is the maximum depth and mtry is the number of sampled dimensions.
> - BNN is unclear: Thanks for this point, we will endeavour to give a better explanation. The reason we didn't include it in the related work is we thought it comparatively simplistic (and perhaps obvious given the rest of the paper content), and due to space restrictions omitted it. We will clarify with the AE on page limit.
> - Cannings and Samworth (2017): This was excluded due to computing limitations. The paper itself uses only small $n$ data sets, and comparatively very few. Although their package offers parallelisation, the compute time is still prohibitive for a reasonable number of bags and a reasonable number of random projections per bag. We do not want to unduly report poor performance because we could not run the method with the intended settings.
> - Scatter plots of pairwise performance: Nice idea, thank you for the suggestion. We will either include these in the main paper (if we can justify the "long paper" format) or the appendix
> - Default parameters: Absolutely, we in fact had these in previously and decided to cut them. We will add them back, along with the performance and in comparison with RF defaults.
>
> Thanks again for all your comments! We will, of course, also attend to your minor comments in our revision.
>
> Best,
>
> authors

---

> > ### Comment · Reviewer_yi1y · 2025-08-29
> >
> > Sounds good, I am looking forward to reading the updated paper. If possible, please upload a version with tracked changes (e.g. via latexdiff) when you submit a revision.
> >
> > > Your comment about the dependence of the subspace on only the $k$-th neighbour (unlike the classifier) is a good one. [...] in fact the difference between using only the $k$-th and all up to the $k$-th, in terms of overall quality of the solutions, is very minimal. If you believe the method would be more "elegant", we would be happy to make the change.
> >
> > I am fine with you keeping the current approach, I would just suggest to add a couple of sentences explaining what you wrote here (and saying that the practical difference is minimal).
> >
> > > as far as we are aware there is a page limit of 12, which is why we moved the details of the classifiers and how we did hyperparameter tuning to the appendix. There is a 16 page "variant" but this requires justification for why we need a "long paper" format.
> >
> > Hmm. Are you sure? This page https://jmlr.org/tmlr/author-guide.html says "Submissions may be any length, but a paper’s length should be justified by its content and papers that are unusually long (not counting any Appendices) are likely to result in reviewing delays." AFAIK, if the paper is below 12 pages, then the reviewers have to submit reviews within 2 weeks (https://jmlr.org/tmlr/reviewer-guide.html), otherwise they get 4 weeks. I am not aware of anything related to 16 pages or any justifications. Where did you get this information from?
> >
> > > we are not professional programmers, whereas the faster RF implementations are hyper-efficient. Actual absolute run-times for our implementation of BOPNN may not be that relevant. We are happy to include them if you think they are instructive. Perhaps better would be something about computational complexity [...]
> >
> > I would definitely suggest to discuss comp. complexity, but I also do think that the actual runtimes are interesting, to give a feeling to the reader. You can of course discuss that the implementation can be further optimized.

---

> > > ### Comment · Reviewer_yi1y · 2025-09-22
> > >
> > > Dear authors, I have now gone over the revision, and it looks good. I will recommend acceptance. Two small things:
> > >
> > > * Are you sure you don't want to reformulate the title from question ("Bags of projected nearest neighbours: competitors with random forests?") into a statement: "Bags of projected nearest neighbours are competitive with random forests" (or "can be competitive")?
> > >
> > > * The phrase "computationally efficient to compute" appears in the abstract and in the introduction; it sounds cumbersome, maybe reformulate.

---

> > > > ### Author Response · Authors · 2025-09-25
> > > > **Thanks for advice**
> > > >
> > > > Dear reviewer,
> > > >
> > > > thanks for the advice on the title. To be honest, we're not entirely sure which we prefer. For some reason the we find the "question format" of the title more "enticing", but, as it would happen, in another paper we had similar advice and so it is very possible that our preference for this style of titles is not held by many. We will put it to the other reviewers, and see if any is willing to comment. Even if not, we're certainly very happy to change it.
> > > >
> > > > We'll definitely rephrase the "computational... compute..." sentence; you're totally right it is clumsy.

---

### Review · Reviewer_oaht · 2025-08-22

**Summary Of Contributions:**

Summary:

This paper proposes BOPNN (Bag Of Projected Nearest Neighbors), a novel approach that integrates adaptive discriminant subspaces into bagged kNN classifiers. By projecting data into subspaces that enhance class separation, BOPNN improves the discriminatory power of kNN ensembles while maintaining interpretability. The authors provide extensive experiments on 162 datasets, showing that BOPNN performs comparably to Random Forests and aligns well with SVM performance in certain settings. The framework also allows for visualization of class structure and assessment of variable importance.

Strengths:

- Introduces a computationally feasible method for adaptive kNN within bagging.

- Extensive empirical evaluation on a large and diverse set of datasets.

- Provides interpretable visualizations and insights into variable importance.

- Demonstrates competitive performance with standard baselines (RF, SVM).

Weaknesses:

- No code release; reproducibility may be limited.

- As mentioned in the paper, performance degrades on datasets with high class imbalance or many categorical variables.

- Requires tuning multiple hyperparameters (k, subspace size, number of eigenvectors).

- Visualization aggregates subspaces, which may obscure individual discriminant directions in the ensemble.

**Audience:**

Yes

**Audience Explanation:**

The paper introduces a novel bagged adaptive kNN approach (BOPNN) that performs comparably to Random Forests and SVMs in many cases, which would be of interest to researchers and practitioners in machine learning, especially those working on ensemble methods and nearest-neighbor models.

**Claims And Evidence:**

Yes

**Claims Explanation:**

The paper provides convincing empirical evidence through extensive experiments on 162 datasets, including comparisons with multiple baseline methods (RF, SVM, etc.) and detailed visualizations. The claims about BOPNN’s performance relative to other methods are supported by charts, tables, and statistical tests.

However, since no code or implementation details are released, the results cannot be independently verified, which slightly weakens the reproducibility aspect.

**Requested Changes:**

Critical changes:
- Code or Reproducibility Materials: Provide code or detailed instructions to reproduce all experiments.
- Clarification on Limitations: Expand discussion on performance with imbalanced classes and high proportions of categorical variables.

Other suggestions:
- Parameter Sensitivity & Computational Cost: Include analysis of hyperparameter effects and training/prediction time relative to Random Forests.
- Additional Visualizations: Show class separation and ensemble projections for datasets with differing performance.

---

> ### Author Response · Authors · 2025-08-24
> **Some initial responses**
>
> Dear reviewer,
>
> thanks very much for your comments! In order to best address your and the other reviewers' comments, we hope to first clarify a few matters and provide some early responses, before conducting any major work towards revisions and additional experiments.
>
> 1. Code release: We have developed an R package, and only did not include this as we were not sure how to ensure anonymity. We will find out how. In fact there is most likely a way we can add supplemental files, including the compressed package, to this submission. But whatever the case, this package will be released along with publication of this work, and we'd appreciate it if you could clarify whether this would satisfy your concerns or if you'd like for us to try and add it as a supplemental file.
> 2. Imbalance and categorical variables: We can absolutely provide some more discussion.
> 3. Parameter sensitivity and computational cost: As mentioned in relation to other reviewers' comments, we are not professional programmers, and as a result our implementation is not nearly as fast as the super-optimised RF implementations. We would argue that computational complexity is the key matter at play. Nonetheless we can include some comparisons for your interest, if you would like.
> 4. Additional visualisations: Nice idea; we will include some in the appendix.

---

> > ### Comment · Reviewer_oaht · 2025-09-09
> >
> > Thanks for the clarifications! Having the R package is great, if there’s a way to include it as a supplemental file for review, that would be ideal, but I understand if it has to wait until publication. Expanding the discussion on imbalanced classes and categorical variables, adding extra visualizations, and including some notes on parameter sensitivity and computational cost will all make the paper stronger.

---

> > > ### Author Response · Authors · 2025-09-10
> > > **Already uploaded revision**
> > >
> > > Dear reviewer,
> > >
> > > as our deadline for revision already passed we submitted a revised version already. If you feel strongly on any points that have not been addressed yet please do let us know.

---

### Review · Reviewer_HiVK · 2025-08-23

**Summary Of Contributions:**

# Summary
The paper proposes BOPNN, a new bagging approach based on kNN classifiers. For each bootstrap sample, the method applies an adaptive learning framework using discriminant subspaces. The authors claim that this could increase the diversity of the base learners without reducing their individual performance. Extensive experiments on 162 datasets show that BOPNN performs on average comparably to Random Forests.

＃Strength
* The experimental evaluation is large-scale and thorough, covering a wide range of benchmark datasets.
* Results indicate that BOPNN performs on par with Random Forests, which are widely considered strong general-purpose classifiers. This finding is potentially valuable for practitioners.

# Weakness
* The validity of the core idea itself, the adaptive learning step via discriminant subspaces, is not sufficiently validated. Although the final classification performance has been verified by many benchmark problems, there are no quantitative verification that the proposed method directly solves the problems pointed out in the introduction. In addition to testing the final performance, it would be useful to examine how the proposed procedure affects the base learners. In particular, it should be shown that the procedure does not reduce the accuracy of each weak learner, and how it increases variance across weak learners. Controlled experiments with synthetic data could help to make this clear, and would be valuable for future work.
* The writing is complex and hard to follow at least for non-natives.  Many sentences are long and contain multiple clauses and modifiers. This structure makes it difficult to follow the argument. Shorter and simpler sentences would improve clarity.

**Audience:**

Yes

**Audience Explanation:**

Ensemble learning and kNN are fundamental techniques. New approaches to improve them are likely to attract interest from readers.

**Broader Impact Concerns:**

This work concerns a potential improvement to a basic algorithm. It is necessary to discuss broad impact.

**Claims And Evidence:**

No

**Claims Explanation:**

As noted in the "Weakness" section, the paper does not directly test whether the proposed idea actually solves the problems of bagging for kNN that were described in the Introduction. In a method paper, overall performance is important, but it is equally important to provide direct evidence that the key idea itself is effective.

**Requested Changes:**

* Please add a section that directly and quantitatively tests the proposed idea. For example, how does the procedure affect the accuracy of the weak learners? How does it change their variance? Controlled experiments, possibly with synthetic data, would make this clear.

* Please improve the writing style. Avoid long sentences with many clauses, and aim for shorter, simpler, and clearer descriptions.

---

> ### Author Response · Authors · 2025-08-24
> **Some initial responses**
>
> Dear reviewer,
>
> thanks very much for your comments, we really appreciate your work! In order to best address your and the other reviewers' comments, we would like to first clarify a few matters and provide some early responses, before conducting any major work towards revisions and, if needed, additional experiments.
>
> 1. Validation of claims: Thanks for this very important comment. We had envisaged that the improvements in performance, ultimately of BOPNN compared with all other bagged NN models, were sufficient to validate these claims, however we acknowledge this is not necessarily a validation of each claim in isolation. To clarify, quickly, there *is* a potential decrease in the accuracy of weak learners when aggressive adaptive learning is conducted, as it increases their variance compared with the passive/lazy alternatives. The combination of the aggressive/adaptive learning *with* the variance reduction of the aggregated model, through bagging, is where the overall benefit lies. In light of this, would you be able to say if this clarifies the matter for you? Specifically, would you like for us to conduct experiments documenting the accuracy of and correlation between the weak learners (simple NN, random subspace NN, adpative subspace NN, and random subspace + adaptive subspace), or do you think some further discussion in the text which clarifies would be preferable?
> 2. Complex writing style: Thank you for pointing this out, as it is an important point. Although one of the other reviewers seemed to particularly like the style in which we have written, we will endeavour to simplify the sentence structure while maintaining overall style.
> 3. Broad impact concerns: Thanks for pointing out that you believe some discussion here is needed. I guess that because our method seems not to perform as well in the context of class imbalance and many categorical variables, its application in the social sciences and where class imbalance may be linked to potential predictive bias relative to certain groups is relevant. Is this the sort of thing you had in mind?

---

> > ### Comment · Reviewer_HiVK · 2025-08-25
> > **A quick response to the broad impact concern**
> >
> > > Broad impact concerns: Thanks for pointing out that you believe some discussion here is needed. I guess that because our method seems not to perform as well in the context of class imbalance and many categorical variables, its application in the social sciences and where class imbalance may be linked to potential predictive bias relative to certain groups is relevant. Is this the sort of thing you had in mind?
> >
> > This is my mistake. The sentence should be
> > > It is __not__ necessary to discuss broad impact.
> > I apologize for this.
> >
> > I will reply again on other points.

---

> > > ### Author Response · Authors · 2025-08-26
> > > **Thanks for clarifying**
> > >
> > > Thank you for taking the time to clarify on this

---

> ### Comment · Reviewer_HiVK · 2025-08-28
> **Regarding the first point**
>
> I agree that the key idea is the combination of adaptiveness and variance reduction by bagging. From the final performance, probably this is what improved the results.
>
> What I am concerned about is whether the procedure really achieves this mechanism directly. Would it be possible to provide quantitative experiments to test this? That would be the desirable analysis.

---

> > ### Author Response · Authors · 2025-08-28
> > **Thanks for the engagement**
> >
> > We believe that the differences in performance: BOPNN > BOpNN > BNN quite strongly indicate that each component (randomness and adaptiveness) improves performance. As to whether "this is achieved", could you possibly clarify what you mean by "this"? Certainly we have included adaptive learning within knn models, and certainly we have added a randomised component, so those are obviously "achieved", and the overall performance is improved. Apologies if we are not understanding your meaning

---

### Review · Reviewer_RcFn · 2025-08-24

**Summary Of Contributions:**

The paper proposes BOPNN, bags of projected nearest neighbors classifier, for improving the performance of kNN with bagging. The authors formulate a k-nearest neighbor covariance matrix, which is the covariance matrix of sample x to its k-th NN, and find the projection direction that maximizes the within-class covariance and minimizes the inter-class covariance.

Strength:
1. The idea is simple and straightforward.
2. The paper conducted quite thorough experiments on 162 ML datasets. The empirical evaluation is very solid to support the claims.

Weakness:
1. The paper lacks theoretical results. It is mostly am empirical paper.
2. As an empirical paper, the experiments should be more comprehensive. I appreciate that the authors tested a lot of datasets, which is impressive. However, more ablation study is needed to better understand what's going on.

For example, the core idea for the proposed method is to "destabalize" individual kNN classifiers. This is done by projecting original data on lower dimensional subspace with a similar idea to LDA or RDA (here the paper uses k-NN covariance matrices). As such, individual learners should be weaker, right? It would be interesting to see the performance of individual classifiers, too.

Also, there should be more detailed breakdown of the results. What's the impact on parameter k? You set k = 1,2,3,4,5, but what's the best choice? How about combining them together instead of just using a single k? This is an important aspect to understand the benefit of proposed metrics in (3).

3. Are there more competing methods in this kind? I assume most dimensionality reduction methods can also be used here, like random projections or PCA, or just LDA/QDA? Can we also compare with these?

**Additional Comments:**

NA

**Audience:**

Yes

**Audience Explanation:**

Bagging and random forest are popular ML tools, so many people will be interested.

**Claims And Evidence:**

No

**Claims Explanation:**

See comments above.

**Requested Changes:**

I would like to see discussions/changes/additional experiments related to my questions and suggestions above.

---

> ### Author Response · Authors · 2025-08-24
> **Some initial responses**
>
> Dear reviewer,
>
> thanks very much for your comments! In an effort to efficiently address the comments of all reviewers, we are hoping to first provide some initial responses and clarifications.
>
> 1. Absolutely we have not attempted to provide any theoretical guarantees associated with our approach. However, we would like to contest that this makes it mostly an empirical piece of work. We believe that there should be more scope in the literature for discussions of fundamental ideas, without necessitating explicit theoretical results. Before continuing, we do not mean to suggest that our approach is somehow fundamental, but rather we believe the success of RFs is somewhat misunderstood by many, and has led to a considerable amount of work being applied to these purely randomised variants of existing models for use within bagged ensembles.  As we mention in the paper, the randomisation can obviously help (and is evidenced by the success of RFs over bags of DTs), but in isolation it is, we believe, never going to suddenly produce something competitive with RFs across many settings. It is the combination of randomness and adaptiveness which is, we believe, the key to RFs success. Having said this, of course without explicit theoretical results we have only been able to support this perspective with experiments, and we apologise if we misunderstood this comment of yours.
>
> 2. We appreciate your comment, and will add some more experiments to try and isolate where the improvements lie. We attempted to do this by including BOpNN (where only the randomised subspace is used, and not the combination of random and adaptive) as well as BNN (no subspace at all) in our experiments. Nonetheless, there are very many variants we could explore, and you're totally correct that other adaptive subspaces will likely work. However, we do believe, and plan to elaborate more on this point, that the particular subspace we use will be superior to many others due to its own inherent high variance. We tried to make this point in Remark 3, but will perform some experiments to document this fact better. Due to page limits, this will need to be an appendix, however.
>
> 3. The point about the weak learners necessarily being weaker will, however, be context dependent. Where the additional model variance exceeds the potential reduction in bias (by filtering out dimensions which do not seem to contributed, based on the sample, to class discrimination) this would indeed be the case, but we are not sure whether this point can be made across the board, nor whether experiments can easily be designed to reflect these facts nor whether they are relevant or not.
>
> 4. With respect to the effect of the different hyperparameters, we would contest that this is arguably tangential to our work. The effect of k on all kNN models is the same. Large k means more smoothing, regardless of whether there is adaptive learning or not, and hence lower variance and higher bias. Using CV to select appropriately is universally supported, we would argue. However, we do believe performance for some default settings is relevant. Regarding having variable values for k within the same ensemble, would not this comment equally apply to RFs, and indeed any and all bagged ensembles? If it is a point about which you feel strongly, we can certainly include experiments documenting the results, however we believe this would be tangential to our main points.

---

> > ### Comment · Reviewer_RcFn · 2025-08-27
> > **Reply to initial response**
> >
> > Thanks for the reply. Let me make my point more clear.
> >
> > The quantity you try to optimize, equation (3), is definitely the core component of the proposed method. My main concern is that this key formula is not well elaborated. The paper did not provide much discussion and evidence on how and why it works. For example
> >
> > 1. As I mentioned, there are many more subspace algorithms. The proposed formulation should be compared with those to justify it is indeed a better metric to optimize.
> >
> > 2. As I also mentioned, regarding the metric itself, the impact of parameter k should also be studied in more details. This is also useful for selecting the initial or default parameter. In computational intensive tasks, this would be very helpful.

---

> > > ### Author Response · Authors · 2025-08-28
> > > **Thanks for engagement**
> > >
> > > We can absolutely include more experiments related to these points. To clarify, you're particularly interested in whether the discriminant matrix formulation we use is actually needed, or if we could just have used LDA/PCA or similar? We can include a section in the appendix where we compare with other subspaces. If you have any others in mind, we'd appreciate the input. MDA is an obvious alternative, but is a bit slower computationally and running all of these across all train/test splits and all 162 data sets may be challenging given other work we have going on. Would running this only over a selection of the data sets be satisfactory? We could focus on some instances where BOPNN does well (answering whether it is because of the subspace we use or just that any random + adaptive + bagging strategy would have) and some where it does poorly (answering whether we could have done better with the same overall model formulation)?
> > >
> > > Regarding choice of k, of course this is relatively straightforward to do, but as there are two other hyper-parameters we would still need to tune over these. Or, would you be satisfied with default settings of these dimensionality parameters and all settings of k?
> > >
> > > Thanks again for the comments and engagement

---

> > > > ### Comment · Reviewer_RcFn · 2025-08-28
> > > > **Reply**
> > > >
> > > > Since the datasets are mostly quite small (at most 10000 samples), I assume PCA of LDA will not be "that slow"? anyways, I'm also fine with a subset of datasets, but of course using all datasets would be more convincing.
> > > >
> > > > For the ablation study on k, I would suggest to report results with the other two parameters fixed at some proper values.

---

> > > > > ### Author Response · Authors · 2025-08-28
> > > > > **Thanks for clarification/confirmation**
> > > > >
> > > > > Thanks a lot for the quick response and further engagement. Absolutely, if it is just PCA and LDA, and not something like MDA, then this can easily be done.

---

### Author Response · Authors · 2025-09-10
**R package**

Dear reviewers,

unfortunately the .tar.gz format is not permissible as supplemental material, so here is a link to the package: https://github.com/code-sharing-anonymised/BOPNN.

Thanks,

authors

---

### Author Response · Authors · 2025-09-25
**Change the title of the paper?**

Dear reviewers,

reviewer yi1y has suggested a change of the title of the paper to "Bags of projected nearest neighbours are competitive with random forests". If you're willing to comment on preference, regardless of whether you recommend the paper for acceptance or not, we would be very appreicative.

With thanks,

authors

---

### Decision · Action_Editor_nDt5 · 2025-10-08

**Recommendation:** Accept as is

**Audience:**

Yes

**Audience Explanation:**

The work offers a simple, interpretable, and empirically strong alternative to Random Forests using adaptive kNN ensembles, a topic of clear interest to TMLR readers focused on ensemble methods, non-parametric learning, and practical algorithm design.

**Claims And Evidence:**

Yes

**Claims Explanation:**

Extensive experiments on 162 datasets, competitive baselines (RF, SVM), ablations (subspace variants, defaults/k), statistical tests, and a clarified training procedure substantiate the claims; despite limited theory and no full variance-decomposition, the empirical evidence is convincing.